# Cholinergic modulation of hippocampal calcium activity across the sleep-wake cycle

Heng Zhou, Kevin R Neville, Nitsan Goldstein, Shushi Kabu, Naila Kausar, Rong Ye, Thuan Tinh Nguyen, Noah Gelwan, Bradley T Hyman, Stephen N Gomperts*

MasGeneral Institute for Neurodegenerative Disease, Department of Neurology, Massachusetts General Hospital, Charlestown, United States

**Abstract** Calcium is a critical second messenger in neurons that contributes to learning and memory, but how the coordination of action potentials of neuronal ensembles with the hippocampal local field potential (LFP) is reflected in dynamic calcium activity remains unclear. Here, we recorded hippocampal calcium activity with endoscopic imaging of the genetically encoded fluorophore GCaMP6 with concomitant LFP in freely behaving mice. Dynamic calcium activity was greater in exploratory behavior and REM sleep than in quiet wakefulness and slow wave sleep, behavioral states that differ with respect to theta and septal cholinergic activity, and modulated at sharp wave ripples (SWRs). Chemogenetic activation of septal cholinergic neurons expressing the excitatory hM3Dq DREADD increased calcium activity and reduced SWRs. Furthermore, inhibition of muscarinic acetylcholine receptors (mAChRs) reduced calcium activity while increasing SWRs. These results demonstrate that hippocampal dynamic calcium activity depends on behavioral and theta state as well as endogenous mAChR activation.
DOI: https://doi.org/10.7554/eLife.39777.001

*For correspondence:
sgomperts@partners.org

Competing interests: The authors declare that no competing interests exist.

## Introduction

Prominent oscillations in the hippocampal local field potential (LFP) entrain neuronal activity across behavioral states. The theta rhythm, a 5–12 Hz oscillation dependent on cholinergic and GABAergic inputs from the medial septum, predominates during exploratory behavior and characterizes REM sleep (*Winson, 1978*; *Rawlins et al., 1979*; *Lawson and Bland, 1993*; *Zhang et al., 2010*; *Buzsáki, 2005*). In contrast, during quiet wakefulness and slow wave sleep, states of low cholinergic tone (*Kametani and Kawamura, 1990*; *Marrosu et al., 1995*; *Buzsáki et al., 1983*; *Zylla et al., 2013*), the LFP transitions to irregular activity punctuated by sharp wave-ripples (SWRs), brief, high frequency oscillations in the local field potential associated with bursts of neuronal activity (*Buzsáki et al., 1983*; *Buzsáki, 2015*).

Recently, large scale imaging of neuronal somatic calcium ($Ca^{2+}$) activity in behaving animals has been employed as a proxy for electrical recording of action potentials (*Dombeck et al., 2007*; *Dombeck et al., 2010*; *Ziv et al., 2013*; *Sheffield and Dombeck, 2015*; *Hamel et al., 2015*). However, the impact of these behavioral and electrophysiologic states on dynamic $Ca^{2+}$ activity is unknown. The tight relation between single action potentials and $Ca^{2+}$ events in reduced preparations (*Ohkura et al., 2012*) supports the expectation that $Ca^{2+}$ events, like action potentials, would coordinate with theta and SWRs. However, neuronal $Ca^{2+}$ levels can be regulated by modification of ionic conductances and by release of $Ca^{2+}$ from intracellular stores, the latter a process dependent on InsP3 and ryanodine receptor signaling cascades (*Stutzmann and Mattson, 2011*). As both of these mechanisms are subject to modulation, these observations raise the possibility that the

coordination of hippocampal $Ca^{2+}$ activity with LFP oscillations may depend on neuromodulators such as acetylcholine (*Lewis and Shute, 1967*; *Teles-Grilo Ruivo and Mellor, 2013*).

Medial septal cholinergic neurons densely innervate the hippocampus (*Mizumori et al., 1990*) where they play a key role in hippocampal function (*Teles-Grilo Ruivo and Mellor, 2013*). Their ablation significantly impairs hippocampal dependent learning and memory, and this effect is recapitulated with muscarinic acetylcholine receptor (mAChR) blockade (*Winson, 1978*; *Mizumori et al., 1990*; *Hasselmo, 2006*). ACh has many actions in the hippocampus. It contributes to hippocampal theta, reduces SWR activity (*Vandecasteele et al., 2014*; *Norimoto et al., 2012*), and selectively enhances entorhinal cortex afferents relative to those from CA3 (*Hasselmo, 2006*). At the single neuron level, ACh depolarizes pyramidal cells and reduces both spike frequency adaptation and the slow afterhyperpolarization (*Teles-Grilo Ruivo and Mellor, 2013*; *Hasselmo, 2006*; *Giocomo and Hasselmo, 2005*; *Brown and Adams, 1980*; *Fernández de Sevilla and Buño, 2003*; *Cole and Nicoll, 1983*). In addition, via a muscarinic mechanism, ACh has been shown to increase the rate of large somatic $Ca^{2+}$ transients via release from InsP3-sensitive $Ca^{2+}$ stores (*Power and Sah, 2002*; *Cho et al., 2008*) and to uncouple $Ca^{2+}$ from the $Ca^{2+}$ dependent $K^+$ conductance (*Müller and Connor, 1991*). These observations suggest that physiologic changes in cholinergic activity, across the sleep-wake cycle and across LFP states, might have profound effects on dynamic $Ca^{2+}$ activity.

To determine the relation between hippocampal physiologic state and $Ca^{2+}$ activity across behavioral states and the sleep-wake cycle, here we imaged dynamic $Ca^{2+}$ activity in neurons of the CA1 pyramidal cell layer and acquired concomitant LFP recordings as mice performed a spatial task and subsequently slept. To evaluate a causal role for medial septal cholinergic neurons, we blocked mAChRs pharmacologically and drove ACh release via selective expression and activation of the excitatory DREADD hM3Dq (*Roth, 2016*) in medial septal cholinergic neurons. We found that neuronal $Ca^{2+}$ activity was markedly more robust during theta- and ACh-associated exploratory behavior and REM sleep compared to quiet wakefulness and slow wave sleep. In addition, somatic $Ca^{2+}$ activity was strongly modulated around SWR events. Further, hM3Dq-mediated activation of medial septal ACh neurons increased dynamic $Ca^{2+}$ activity while reducing SWRs, and somatic $Ca^{2+}$ activity - in both wildtype animals and in chemogenetically activated animals expressing hM3Dq in septal cholinergic neurons - was markedly suppressed with the muscarinic AChR (mAChR) antagonist scopolamine. Together, these results demonstrate that hippocampal dynamic calcium activity depends on behavioral and LFP state as well as on endogenous mAChR activation.

## Results

### Coordination of hippocampal dynamic calcium activity across behavior

We imaged somatic $Ca^{2+}$ activity from CA1 neurons and simultaneously recorded LFP as mice traversed a linear track and subsequently rested quietly and slept in a post-behavioral recording session (n = 5 mice, cells/session: 88, 119, 130, 134, 171; *Figure 1A–H*; *Figure 1—figure supplement 1*). There was a significant main effect of behavioral state on $Ca^{2+}$ event rates (repeated measures ANOVA: $F_{(3, 12)}=17.97$, p<0.001; *Figure 1I*). $Ca^{2+}$ event rates were markedly higher in run behavior (>3 cm/sec) than in all other states and were lowest in SWS (post-hoc comparisons with run: quiet wakefulness, p=0.007; slow wave sleep, p=0.003; REM, p=0.011; *Figure 1I* and *Video 1*). Interestingly, with the advent of REM sleep, $Ca^{2+}$ activity increased from slow wave sleep to levels approaching those observed during wakefulness (REM *vs.* slow wave sleep, p=0.050; *Figure 1I* and *Video 1*). There was also a significant main effect of behavioral state on $Ca^{2+}$ event amplitudes (repeated measures ANOVA: $F_{(3, 12)}=4.488$, p=0.025; *Figure 1J*). In post-hoc comparisons, $Ca^{2+}$ amplitudes were significantly smaller in slow wave sleep than in run behavior (p=0.042) or quiet wakefulness (p=0.032). Multiunit activity was not markedly modulated across behavioral states in these recordings (repeated measures ANOVA: $F_{(2, 6)}=0.591$, p=0.544; *Figure 1K*), although a microscope-associated electrical noise artifact prevented contrasts of run behavior with other behavioral states. As expected, there was a significant main effect of behavioral state on theta power (repeated measures ANOVA excluding REM: $F_{(2,8)}=20.468$, p=0.001), with higher theta power in exploratory behavior than quiet wakefulness and SWS (post hoc comparisons with run: quiet wakefulness, p=0.003; slow wave sleep, p=0.015; *Figure 1L*). Distributions of instantaneous $Ca^{2+}$ event rates often had a small secondary peak above 0.5 Hz (*Figure 1M*). The majority of these $Ca^{2+}$ event

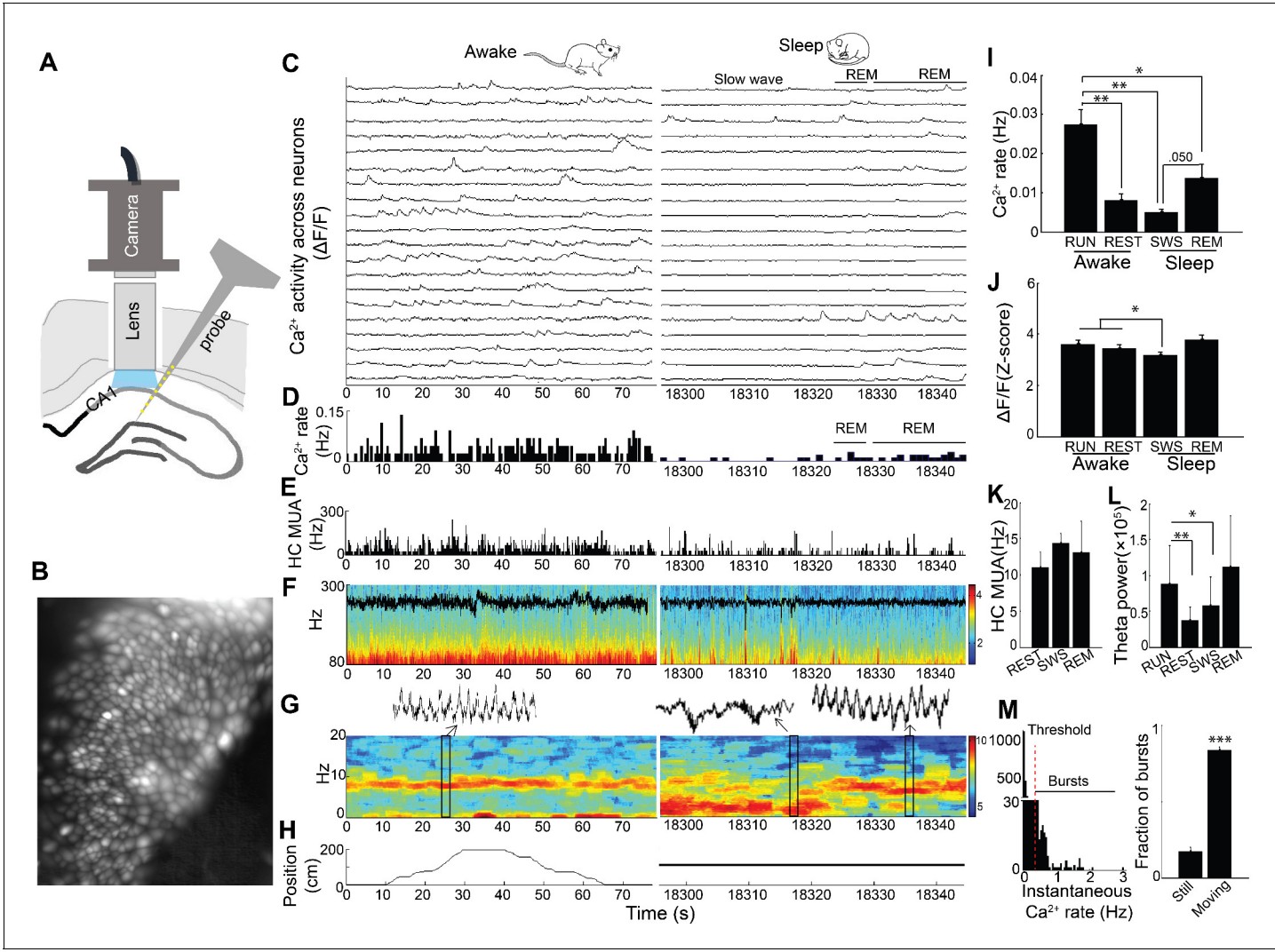

**Figure 1.** Simultaneous dynamic calcium imaging and local field potential (LFP) recording in hippocampus across behavioral states. (A) Schematic. (B) Identified CA1 neurons for calcium imaging in a single freely behaving mouse. (C) Calcium activity of imaged neurons (20/130 neurons of one mouse are shown) across Awake (Left) and Sleep (Right) states. (D) Histogram of $Ca^{2+}$ transient events of all 130 imaged cells (bin size, 500 ms). (E) Histogram of multi-unit activity (bin size, 50 ms). (F) 80–300 Hz spectrograms of LFP recorded in s. pyramidale during running behavior (Left) and sleep (Right), with LFP superimposed. (G) 0–20 Hz spectrograms of LFP recorded in s. radiatum during running behavior (Left) and sleep (Right). Insets: theta in s. radiatum during run behavior (Left, 2 s), two SWRs in s. pyramidale during slow wave sleep (Right, 2 s), theta in s. radiatum during REM sleep (Right, 2 s). (H) The animal's position on the linear track (Left) and during the post-behavioral session (Right) as a function of time. (I) Calcium event rates within neurons were higher during run behavior and REM sleep than during quiet wakefulness (Rest) and slow wave sleep (SWS; n = 5 mice with 88, 119, 130, 134, 171 neurons). (J) Calcium event amplitudes within neurons were lower in SWS than in run behavior or rest. (K) Hippocampal multiunit activity (MUA) across behavioral states. A microscope-associated electrical noise artifact on the track prevented contrasts of multi-unit activity in run with other behavioral states. (L) Theta power is higher in run behavior than quiet wakefulness and SWS (n = 5 mice). As REM was identified on the basis of theta, REM was excluded from statistical contrasts. (M) Bursts of $Ca^{2+}$ events (>0.5 Hz of instantaneous calcium rates) evident in inter-event interval histograms were more prevalent with movement (>0.5 cm/s; n = 6 mice). Statistical comparisons in I–L) were performed using one-way repeated measures ANOVA with *post hoc* tests. That of M) was performed with a paired t-test. *p<0.05, **p<0.01 and ***p<0.001.

DOI: https://doi.org/10.7554/eLife.39777.002

The following source data and figure supplement are available for figure 1:

**Source data 1.**
DOI: https://doi.org/10.7554/eLife.39777.004

**Figure supplement 1.** Representative GCaMP6f virus expression and lens and probe position for simultaneous $Ca^{2+}$ imaging and local field potential recording.
DOI: https://doi.org/10.7554/eLife.39777.003

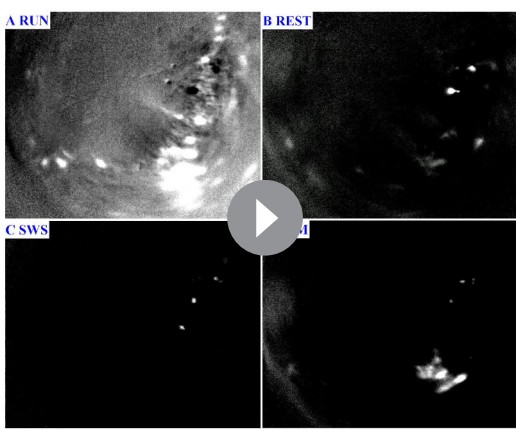

**Video 1.** Representative calcium activity from a single mouse recording across the sleep-wake cycle. SWS (slow wave sleep), REM (rapid eye movement). Time: 20 s.
DOI: https://doi.org/10.7554/eLife.39777.005

bursts, defined using this 0.5 Hz threshold, were movement-associated ($p=6.5\times10^{-5}$, paired t-test; *Figure 1M*). Even so, we did not detect a consistent relationship between animal speed and $Ca^{2+}$ event rates at the individual or group level. Thus, dynamic $Ca^{2+}$ activity varied dramatically with behavior and across the sleep wake cycle, with substantially higher rates of $Ca^{2+}$ events during the high theta conditions of running behavior and REM sleep.

Because hippocampal LFP varies across behavioral states, we next sought to assess the relation of LFP state to neuronal $Ca^{2+}$ activity. In SWR events, synchronized depolarizations drive large populations of CA1 neurons to fire. To determine whether such activity patterns are reflected in the coordination of neuronal $Ca^{2+}$ activity with SWRs, as shown previously (*Malvache et al., 2016*), we evaluated $Ca^{2+}$ activity triggered on SWR events (*Figure 2*). There was a significant main effect of timing relative to SWRs on $Ca^{2+}$ event rates (repeated measures ANOVA: $F_{(2, 14)}=10.802$, p=0.004; n = 8 mice; *Figure 2J*, left). Interestingly, this approach detected a robust reduction in somatic $Ca^{2+}$ activity in association with SWRs, compared to baseline 2–5 s earlier and the immediate 1–2 s post-SWR period (post-hoc comparisons: Pre *vs.* SWR, p=0.022; Post *vs.* SWR, p=0.005; *Figure 2E,F and J*, left). SWRs were not associated with a detectable change in animal movement (speed, mean ± s.e.: Pre 1.7 ± 0.4 cm/sec, SWR 1.5 ± 0.3 cm/sec, Post 1.5 ± 0.4 cm/sec; ANOVA: $F_{(2, 12)}=1.71$, p=0.23). Place cells, cells with $Ca^{2+}$ event place fields on the track, showed a similar reduction of activity around SWR events (*Figure 2—figure supplement 1*). These results demonstrate marked modulation of $Ca^{2+}$ activity around SWRs.

The SWR-triggered reduction in dynamic $Ca^{2+}$ activity started and was often greatest before the SWR began (minimum, mean ± s.e.: −0.075 ± 0.118 s). To refine the temporal relationship between $Ca^{2+}$ activity and SWRs, we first accounted for the delay between $Ca^{2+}$ transient detection, defined as the time at which the event reached 20% of its maximum amplitude, and the start of $Ca^{2+}$ transients on average (0.114 ± 0.001 s; n = 7 mice). Accounting for this delay, the maximal reduction in $Ca^{2+}$ activity preceded SWR onset by 0.189 ± 0.118 s. To enrich for dendritic $Ca^{2+}$ activity capable of reflecting SWR-associated synaptic activity, we took advantage of extra-somatic GCaMP6f expression in dendrites (*Chen et al., 2013*) (*Figure 2—figure supplement 2*). Consistent with the capacity for $Ca^{2+}$ imaging to detect the SWR-associated synaptic barrage with temporal fidelity, SWRs were associated with a time-locked (mean ± s.e.: −0.04 ± 0.10 s) change in the extra-somatic $Ca^{2+}$ fluorescence of the neuropil, with a nonsignificant increase followed by a significant, delayed reduction (repeated measures ANOVA: $F_{(2, 12)}=15.627$, p=0.003, n = 7 animals; post-hoc comparisons: Pre *vs.* SWR, p=0.130; Pre *vs.* Post, p<0.001; SWR *vs.* Post, p=0.003; *Figure 2G and K*, left). To independently assess the temporal alignment of the LFP and the $Ca^{2+}$ fluorescence data, we placed a stimulating electrode in contralateral CA3 to activate the commissural pathway (*Gruart et al., 2015*; *Dong et al., 2008*) and evoke both a field EPSP and an associated change in GCaMP6f fluorescence in CA1 neurons. Stimulation evoked a field EPSP with mean latency 20.8 ± 0.7 ms and an associated increase in somatic $Ca^{2+}$ fluorescence that started within 66.7 ± 16.7 ms (1.3 ± 0.3 samples) and peaked within 100 ± 0 ms (2 ± 0 samples; n = 3 animals; *Figure 2—figure supplement 3*). Accounting for this offset and the ~190 ms offset delineated above further advances the timing of dynamic $Ca^{2+}$ activity relative to SWR onset. Together, these results support the observation that dynamic $Ca^{2+}$ activity transiently falls in the immediate pre-SWR period.

The unexpected absence of a detectable increase in $Ca^{2+}$ activity at SWRs led us to consider the possibility that such an increase in $Ca^{2+}$ activity may depend on SWR dynamics. We therefore compared $Ca^{2+}$ activity triggered on single, isolated SWRs to $Ca^{2+}$ activity triggered on trains of discrete SWRs (*Buzsáki et al., 1983*; *Davidson et al., 2009*) anticipated to provide a more effective somatic

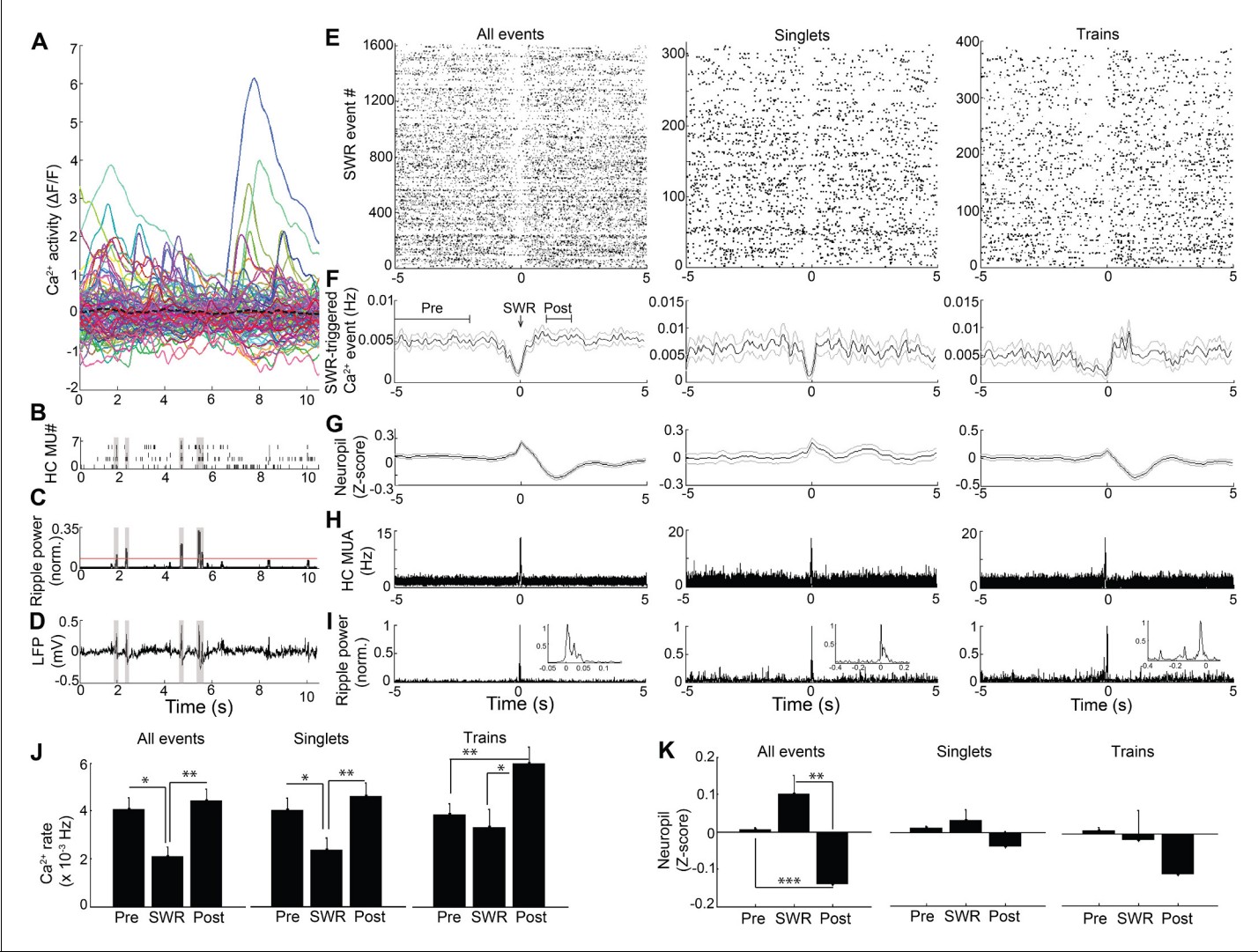

**Figure 2.** Modulation of Ca²⁺transients at SWRs. (A) Temporal correspondence of simultaneously acquired dynamic calcium activity in 130 individual neurons with (B) CA1 multiunit action potentials as well as (C) ripple power and (D) LFP in a single freely behaving mouse. The dashed black line in (A) shows the global mean calcium activity. The red line in (C) reflects the threshold ripple power used to identify SWR events. (E) Raster and (F) peri-event time histogram of calcium events across all SWR events from the same mouse revealed a reduction of dynamic calcium activity that started before SWR onset. Raster and peri-event time histogram of calcium events across SWR singlets and across SWR trains are also shown. (G) Peri-event average neuropil fluorescence reveals SWR-associated calcium dynamics. (H) SWR-triggered multiunit action potentials and (I) pyramidal cell layer ripple power. Inset, expanded timescale. (J) Across all SWRs and SWR singlets, calcium transient rates at SWR onset (time = 0) were reduced compared to the preceding 2–5 s baseline and the 1–2 s post-SWR period (analysis times shown in F). After SWR trains, calcium rates were increased compared to baseline and SWR onset (n = 7 mice with 88, 112, 119, 130, 134, 171, 222 neurons). (K) Across all SWRs, the change in neuropil fluorescence was diminished in the post-SWR period (analysis times shown in (F); n = 6 mice). All events, all SWRs; Singlets, Single SWR event (separated by 1 s or more); Trains, SWR trains (doublets, triplets, etc.). (J–K) Statistics were performed using one-way repeated measures ANOVA with *post hoc* tests. *p<0.05, **p<0.01.

DOI: https://doi.org/10.7554/eLife.39777.006

The following source data and figure supplements are available for figure 2:

**Source data 1.**

DOI: https://doi.org/10.7554/eLife.39777.011

**Figure supplement 1.** Calcium activity of place cells at SWR events.

DOI: https://doi.org/10.7554/eLife.39777.007

**Figure supplement 2.** Fluorescence measurement in the neuropil.

DOI: https://doi.org/10.7554/eLife.39777.008

**Figure supplement 3.** Commissural stimulation evoked field EPSP and calcium fluorescence changes.

*Figure 2 continued*

DOI: https://doi.org/10.7554/eLife.39777.010

**Figure supplement 3—source data 1.**

DOI: https://doi.org/10.7554/eLife.39777.009

depolarization. Transient reduction of $Ca^{2+}$ event rates was preserved when triggered on SWR singlets (repeated measures ANOVA: $F_{(2, 14)}$=9.628, p=0.005; post-hoc comparisons: Pre *vs.* SWR, p=0.033; Post *vs.* SWR, p=0.007, n = 8 animals; *Figure 2E,F and J*, Middle), without clear evidence for a subsequent increase in $Ca^{2+}$ event rates (Post *vs.* Pre, p=0.11). In contrast, when selectively triggered on the last SWR in a SWR train, $Ca^{2+}$ event rates transiently increased (repeated measures ANOVA: $F_{(2, 14)}$=9.145, p=0.003; post-hoc comparisons: Pre *vs.* SWR, p=0.447; Pre *vs.* Post, p=0.003; SWR *vs.* Post, p=0.010 n = 8 animals; *Figure 2E,F and J*, Right). SWR singlets and trains showed only trend level fluorescence changes in the neuropil (repeated measures ANOVA: Singlets: $F_{(2, 12)}$=2.242, p=0.149; Trains: $F_{(2, 12)}$=2.984, p=0.089; *Figure 2G and K*, Middle and Right). These results show that while a transient suppression of dynamic $Ca^{2+}$ activity precedes SWRs, trains of temporally clustered SWRs are capable of eliciting a robust increase in dynamic $Ca^{2+}$ activity.

To further evaluate the relationship of SWR trains to $Ca^{2+}$ activity, we also performed an inverse analysis, in which we evaluated the LFP associated with synchronous $Ca^{2+}$ events across neurons. Synchronous events were relatively rare (10 ± 6 per recording, n = 6 animals). Even so, consistent with prior reports (*Malvache et al., 2016*), 12.6 ± 5.6% of synchronous $Ca^{2+}$ events time coincided with SWRs. Compared to temporally shuffled data, 4.9 ± 3.1% of synchronous $Ca^{2+}$ activities was associated with increased ripple power (p<0.05, n = 5 animals). These data confirm the capacity for synchronous $Ca^{2+}$ events to coincide with SWRs.

Thus, the SWR-synchronized depolarization of neurons, which is associated with hippocampal sequence replay and is implicated in memory consolidation (*Buzsáki, 2015*; *Gomperts et al., 2015*), was only associated with robust increases in dynamic $Ca^{2+}$ activity in vivo when SWRs clustered in sets. Together, these results suggest that an underlying association of $Ca^{2+}$ activity with hippocampal theta state and modulation of activity at SWR events contributed to the marked effect of behavioral state on neuronal $Ca^{2+}$ activity.

## Role for acetylcholine in hippocampal neuron dynamic calcium activity

Previous work has shown that medial septum cholinergic neurons critically contribute to the hippocampal theta oscillation during exploratory behavior (*Winson, 1978*; *Rawlins et al., 1979*; *Lawson and Bland, 1993*; *Vandecasteele et al., 2014*) and that ACh can influence neuronal activity and drive release of $Ca^{2+}$ from intracellular stores (*Stutzmann and Mattson, 2011*; *Hasselmo, 2006*; *Giocomo and Hasselmo, 2005*; *Brown and Adams, 1980*; *Fernández de Sevilla and Buño, 2003*; *Cole and Nicoll, 1983*; *Power and Sah, 2002*). We therefore pursued the hypothesis that medial septal cholinergic inputs contribute to hippocampal dynamic $Ca^{2+}$ activity in freely behaving animals. To determine whether medial septal cholinergic neuron activity was sufficient for the behavioral state- and theta- dependence of hippocampal $Ca^{2+}$ activity, we injected AAV-hSyn-DIO-hM3D(Gq)-mCherry into the medial septum of *ChAT*-Cre mice and prepared mice for hippocampal dynamic $Ca^{2+}$ imaging and LFP co-recording (*Figure 3A*; *Figure 3—figure supplement 1*). Immunohistochemistry confirmed selective expression of hM3Dq in cholinergic cells (*Figure 3B,C*). Upon injection of CNO, the hM3Dq ligand, the rate of dynamic $Ca^{2+}$ activity during quiet wakefulness increased compared to injection of vehicle (p=0.005, paired t-test, n = 6; *Figure 3E,H,J* and *Video 2*). In contrast, hM3Dq activation of cholinergic cells did not increase the amplitude of $Ca^{2+}$ events (p=0.677, paired t-test; *Figure 3I,K*) and did not change average hippocampal multiunit activity (p=0.461, paired t-test, n = 5; *Figure 3F,L*). In addition, CNO did not affect movement or other behaviors (p=0.3, paired t-test, n = 5; *Figure 3G,M*, *Figure 3—figure supplement 2*), including running velocity on the track (p=0.572, paired t-test, n = 5) or the proportion of time spent running (p=0.167, paired t-test, n = 5; *Figure 3—figure supplement 3*). CNO was associated with a reduction in the rate of SWRs (p=0.035, one tailed paired t-test; *Figure 3N*), consistent with prior reports (*Vandecasteele et al., 2014*; *Norimoto et al., 2012*), but did not significantly alter hippocampal theta power in quiet wakefulness or exploratory behavior (each contrast, p=0.290, paired

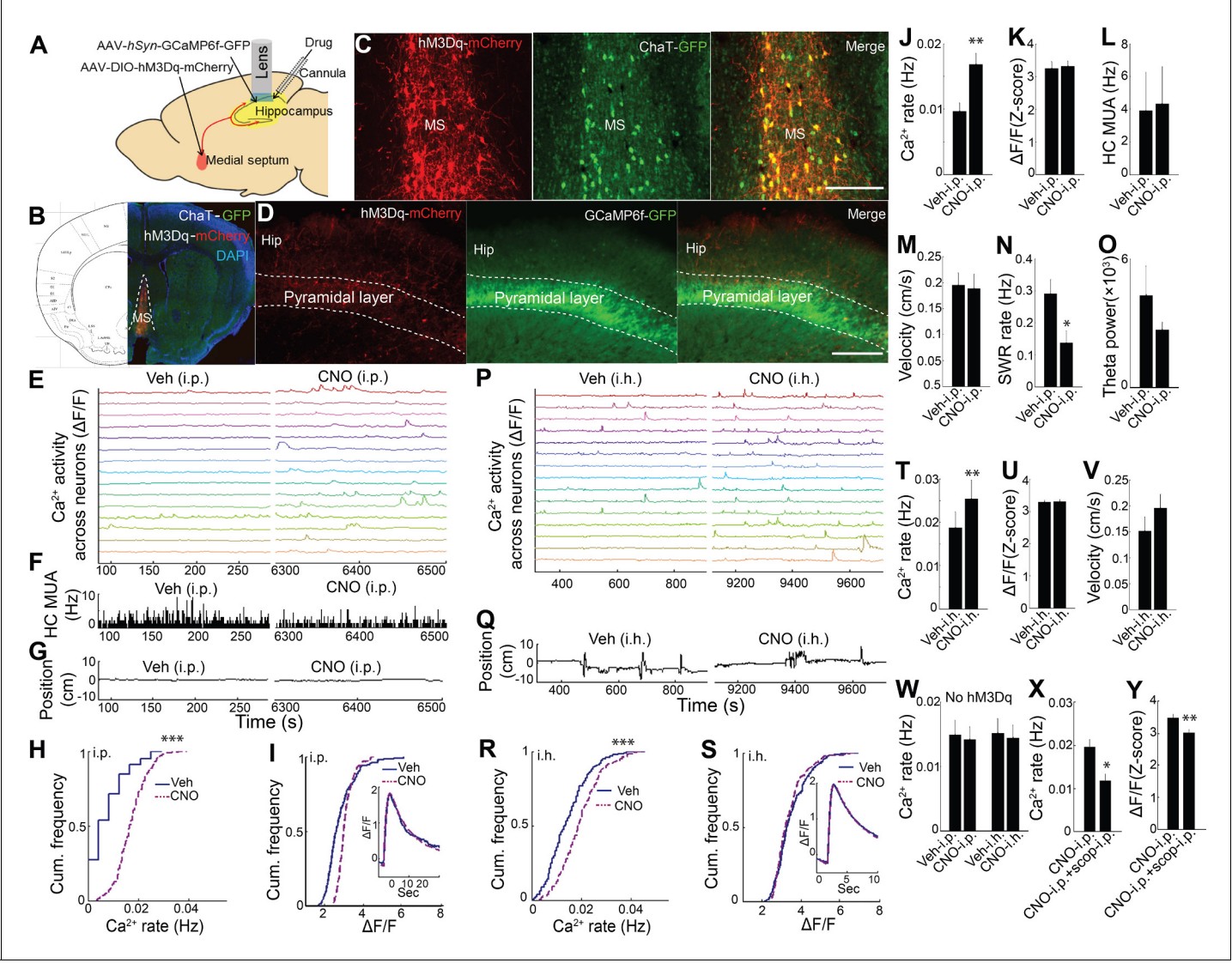

**Figure 3.** Chemogenetic activation of septal cholinergic neurons increased CA1 dynamic calcium activity in ChAT-Cre mice. (**A**) Schematic for local injection. (**B**) Immunohistochemistry shows the expression of hM3Dq in the medial septum (MS) and (**C**) colocalization with cholinergic neurons (yellow): Red, anti-mCherry for hM3Dq; GFP, anti-ChAT. (**D**) The axons of cholinergic neurons expressing hM3Dq were also detected in the hippocampus, where CA1 neurons expressed GCaMP6f (GFP). Calibration bar: 200 µm. (**E**) Systemic (i.p) CNO injection (Right) increased calcium activity in individual neurons compared to vehicle (Left) (15/237 neurons of one mouse are shown). (**F**) Histogram of hippocampal multiunit activity (MUA). (**G**) Animal position during quiet wakefulness. (**H**) CNO injection increased calcium event rates within neurons but not (**I**) calcium event amplitudes. Inset, average Ca²⁺ event waveforms after vehicle and after CNO injection. Group data (n = 6 mice with 65, 98, 119, 184, 212, 237 neurons) show the effects of cholinergic activation with hM3Dq on (**J**) calcium event rates and (**K**) calcium event amplitudes. (**L**) Systemic CNO injection did not change hippocampal MUA (n = 5), (**M**) Velocity in the post-behavioral session after vehicle and CNO (i.p.) injection. (**N**) Systemic CNO injection reduced the rate of SWRs (n = 5 mice). (**O**) CNO injection did not change theta power. (**P**) Intrahippocampal (i.h.) CNO injection (Right) increased calcium activity across individual neurons compared to vehicle (Left) (15/212 neurons of one mouse are shown). (**Q**) Animal position in quiet wakefulness. (**R**) CNO i.h. injection increased calcium event rates within neurons but not (**S**) calcium event amplitudes. Inset, average Ca²⁺ event waveforms after vehicle and after CNO. Group data (n = 5 mice with 94, 113, 206, 212, 250 neurons) show the effects of local CNO injection on (**T**) calcium event rates and (**U**) amplitudes. (**V**) CNO i.h. injection did not affect animal velocity. (**W**) Neither i.p. nor i.h CNO injection affected calcium activity in mice without hM3Dq injection in the medial septum. The effects of CNO i.p. injection on (**X**) calcium rates and (**Y**) amplitudes were blocked with scopolamine. Statistics were performed using paired t-tests, *p<0.05, **p<0.01, ***p<0.001.

DOI: https://doi.org/10.7554/eLife.39777.012

The following source data and figure supplements are available for figure 3:

**Source data 1.**

DOI: https://doi.org/10.7554/eLife.39777.020

*Figure 3 continued*

**Figure supplement 1.** Representative lens and cannula positioning.

DOI: https://doi.org/10.7554/eLife.39777.013

**Figure supplement 2.** CNO systemic injection did not affect other behaviors.

DOI: https://doi.org/10.7554/eLife.39777.014

**Figure supplement 2—source data 1.**

DOI: https://doi.org/10.7554/eLife.39777.015

**Figure supplement 3.** The effects of CNO systemic injection on velocity.

DOI: https://doi.org/10.7554/eLife.39777.016

**Figure supplement 3—source data 1.**

DOI: https://doi.org/10.7554/eLife.39777.017

**Figure supplement 4.** The effect of CNO injection on calcium event amplitudes in mice that did not undergo hM3Dq injection in the medial septum.

DOI: https://doi.org/10.7554/eLife.39777.018

**Figure supplement 4—source data 1.**

DOI: https://doi.org/10.7554/eLife.39777.019

t-tests, n = 5; *Figure 3O*). Neither CNO-associated changes in SWR rates (p=0.4, Spearman) nor changes in theta power (p=0.7, Spearman) significantly correlated with CNO-associated changes in $Ca^{2+}$ event rates. To evaluate for CNO off-target effects (*Gomez et al., 2017*), CNO was injected in animals lacking hM3Dq receptor expression. No effect was observed on the rate of dynamic $Ca^{2+}$ activity ($Ca^{2+}$ event rates, p=0.496; amplitudes, p=0.710; paired t-test, n = 5; *Figure 3W* and *Figure 3—figure supplement 4*). These data show that activation of medial septal cholinergic neurons was sufficient to impact hippocampal LFP state and to increase neuronal $Ca^{2+}$ activity.

We next sought to confirm that these results could be recapitulated via activation of septal cholinergic afferents to the hippocampus. Consistent with prior reports (*Teles-Grilo Ruivo and Mellor, 2013*), after expression of hM3Dq in medial septal cholinergic neurons, hM3Dq-containing axons could be observed in CA1 (*Figure 3D*). We therefore directly injected CNO into the dorsal hippocampus ipsilateral to the microscope while imaging $Ca^{2+}$ activity. Compared to intra-hippocampal injection of vehicle, intra-hippocampal injection of CNO increased the rate of CA1 neuronal $Ca^{2+}$ activity (p=0.008, paired t-test; n = 5; *Figure 3P,R,T* and *Video 2*) without affecting $Ca^{2+}$ event amplitudes (p=0.810, paired t-test; *Figure 3S,U*) or animal behavior during quiet wakefulness (p=0.248, paired t-test; *Figure 3Q,V*). Intra-hippocampal injection of CNO in animals lacking hM3Dq receptor expression (*Gomez et al., 2017*) did not alter on the rate of dynamic $Ca^{2+}$ activity ($Ca^{2+}$ event rates, p=0.637; amplitudes, p=0.128; paired t-test, n = 4; *Figure 3W* and *Figure 3—figure supplement 4*). The effect of hM3Dq activation of cholinergic cells on CA1 $Ca^{2+}$ activity was also mAChR-dependent, as hippocampal neuronal $Ca^{2+}$ activity after hM3Dq activation of cholinergic cells was robustly inhibited with mAChR antagonist scopolamine injection (i.p.) ($Ca^{2+}$ event rates, p=0.014; amplitudes, p=0.001; paired t-tests, n = 5 mice; *Figure 3X,Y*).

To determine whether endogenous muscarinic ACh receptor (mAChR) activation mediated the dependence of dynamic $Ca^{2+}$ activity on behavioral and theta state, we again employed the selective mAChR antagonist scopolamine in WT animals. Scopolamine systemic treatment markedly reduced hippocampal dynamic $Ca^{2+}$ event rates and amplitudes during exploratory behavior (rate: p=0.014; amplitude: p<0.001, paired t-tests, n = 7; *Figure 4A–G* and *Video 3*). In contrast, scopolamine did not diminish average hippocampal multiunit activity (exploratory behavior, p=0.087, paired t-tests, n = 6, *Figure 4H*; quiet wakefulness, p=0.9).

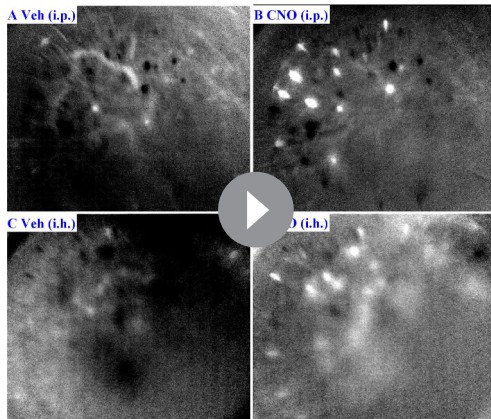

**Video 2.** Representative calcium activity after Vehicle and CNO injections (i.p. or i.h.). Time: 20 s.

DOI: https://doi.org/10.7554/eLife.39777.021

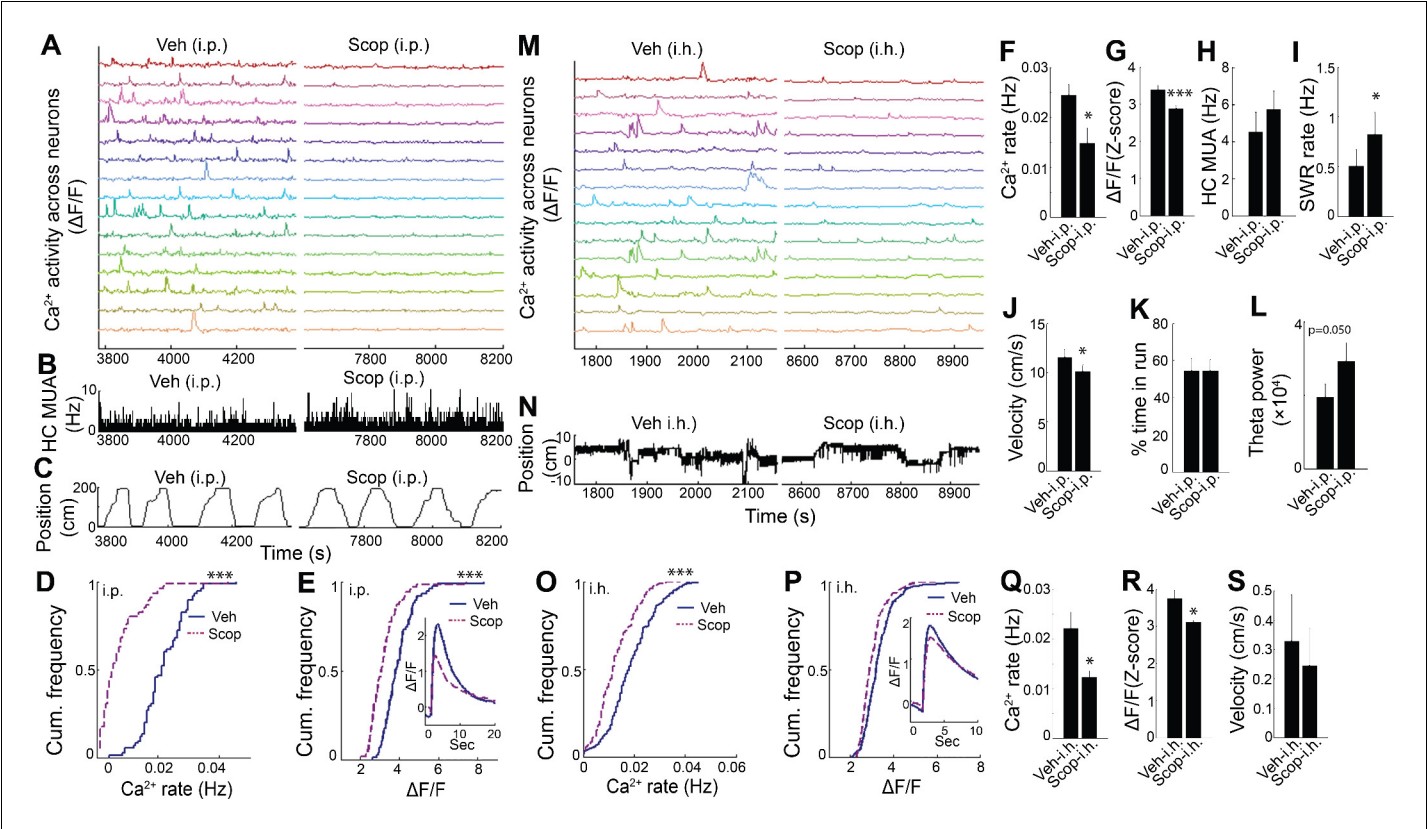

**Figure 4.** Hippocampal dynamic calcium activity was mAChR dependent. (**A**) Systemic (i.p) scopolamine injection (Right) reduced calcium activity across individual neurons compared to vehicle (Left) (15/75 neurons of one mouse are shown). (**B**) Histogram of hippocampal multiunit activity (MUA) after injection of vehicle and scopolamine. (**C**) Animal position as a function of time on the track. Both (**D**) calcium event rates and (**E**) event amplitudes were reduced after scopolamine injection. Inset, average Ca2+ event waveforms after vehicle and after scopolamine. Group level effects of scopolamine on (**F**) calcium event rates and (**G**) event amplitudes during run behavior (n = 7 mice with 73, 74, 75, 106, 183, 219, 225 neurons). (**H**) Scopolamine i.p. did not change hippocampal MUA. (**I**) Scopolamine injection increased SWR rates (n = 6 mice). (**J**) Scopolamine injection reduced run velocity but not (**K**) the percentage of time spent in run on the task. (**L**) Effect of scopolamine injection on hippocampal theta power. (**M**) Intrahippocampal (i.h.) scopolamine injection (Right) reduced calcium activity across individual neurons compared to vehicle (Left) (15/239 neurons of one mouse are shown). (**N**) Animal position during quiet wakefulness. (**O**) Calcium event rates and (**P**) event amplitudes after vehicle and scopolamine injection. Group level effects of i.h. scopolamine injection on (**Q**) calcium event rates and (**R**) event amplitudes during quiet wakefulness (n = 5 mice with 31, 119, 154, 239, 263 neurons). (**S**) Intrahippocampal scopolamine injection did not affect animal activity during quiet wakefulness (n = 5 mice). Statistics were performed using paired t-tests, *p < 0.05, ***p < 0.001.test

DOI: https://doi.org/10.7554/eLife.39777.022

The following source data and figure supplements are available for figure 4:

**Source data 1.**
DOI: https://doi.org/10.7554/eLife.39777.027
**Figure supplement 1.** Scopolamine systemic injections did not affect other behaviors.
DOI: https://doi.org/10.7554/eLife.39777.023
**Figure supplement 1—source data 1.**
DOI: https://doi.org/10.7554/eLife.39777.024
**Figure supplement 2.** The effects of scopolamine systemic injection on velocity.
DOI: https://doi.org/10.7554/eLife.39777.025
**Figure supplement 2—source data 1.**
DOI: https://doi.org/10.7554/eLife.39777.026

Scopolamine also increased SWR event rates during periods of quiet wakefulness (p=0.012, paired t-test; *Figure 4I*) and increased theta power during exploratory behavior (p=0.05, paired t-test, n = 7; *Figure 4L*). Neither scopolamine-associated changes in SWR rates (p=0.3, Spearman) nor changes in theta power (p=0.14, Spearman) significantly correlated with changes in Ca2+ event rates.

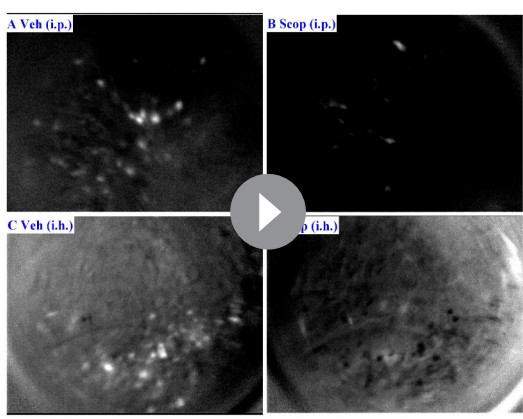

**Video 3.** Representative calcium activity after Vehicle and Scopolamine injections (i.p. or i.h.). Time: 20 s. DOI: https://doi.org/10.7554/eLife.39777.028

Scopolamine did not affect the fraction of time spent running on the track (p=1.0, paired t-test, n = 7; *Figure 4K*) but was associated with a mild reduction in running speed without an effect on other behaviors (p=0.023, paired-test; *Figure 4J*, *Figure 4—figure supplement 1*, and *Figure 4—figure supplement 2*). Thus, the effects of behavioral state and endogenous septal neuron activation on dynamic Ca$^{2+}$ activity were likely mediated by mAChRs.

To confirm that the effect of scopolamine on hippocampal Ca$^{2+}$ activity required inhibition of dorsal CA1 mAChRs, we evaluated the effect of intra-hippocampal injection of scopolamine on dynamic Ca$^{2+}$ activity. Compared to injection of vehicle, intra-hippocampal injection of scopolamine was associated with a significant reduction in both neuronal Ca$^{2+}$ event rates (p=0.043, paired t-test; *Figure 4M,O,Q* and *Video 3*) and amplitudes (p=0.038; *Figure 4P,R*), without significantly affecting behavior (speed in quiet wakefulness, p=0.081; *Figure 4S*). Thus, hippocampal dynamic Ca$^{2+}$ activity was dependent on mACh receptor-mediated signaling in CA1. These data confirm a critical role for medial septal cholinergic neuron-mediated activation of mAChRs in hippocampal dynamic Ca$^{2+}$ activity in freely behaving animals.

## Discussion

We combined dynamic Ca$^{2+}$ imaging of hippocampal CA1 neurons with local LFP recordings in order to relate dynamic Ca$^{2+}$ activity to behavioral and hippocampal state. We found that dynamic Ca$^{2+}$ activity was greatest in exploratory behavior and secondarily in REM sleep, states characterized by strong hippocampal theta oscillations, with marked reductions in quiet wakefulness and slow wave sleep, both non-theta states. Although these findings resemble the state-dependence of pyramidal cell action potential firing rates, which are highest in active wakefulness and REM sleep (*Grosmark et al., 2012*; *Miyawaki and Diba, 2016*), mean Ca$^{2+}$ event rates were comparable in quiet wakefulness and slow wave sleep. In addition, neuronal firing rates appear to change less markedly with state than dynamic Ca$^{2+}$ activity. These results contrast with wide-field calcium imaging in cortex, where Ca$^{2+}$ activity has been shown to fall in SWS compared to wakefulness, consistent with the present results, but with greatest reduction in REM sleep (*Niethard et al., 2016*). Together, these observations suggest that state-dependent and cholinergic effects on dynamic Ca$^{2+}$ activity may not be uniform across brain regions.

Whereas action potentials of CA1 neuronal populations coordinate robustly with SWR events in both wakefulness and sleep (*Buzsáki, 2005*; *Buzsáki, 2015*; *Grosmark et al., 2012*; *Miyawaki and Diba, 2016*), dynamic Ca$^{2+}$ activity of imaged neurons detectably increased only in association with SWR trains and was not evident with SWR singlets or across SWRs on average. While it remains possible that neuronal firing during single SWRs increases cytoplasmic Ca$^{2+}$ levels, perhaps via Ca$^{2+}$ transients that follow LFP-detected SWRs, such activity was insufficiently robust to increase average Ca$^{2+}$ transient rates. This result may derive from the prominent contribution of inhibition during SWRs coupled with potential effects of SWR trains on the post-SWR hyperpolarization (*Gan et al., 2017*; *Hulse et al., 2016*; *English et al., 2014*; *Maier et al., 2011*). These differences in Ca$^{2+}$ activity across SWR singlets and trains may reflect a distinct role for dynamic Ca$^{2+}$ activity in trains, for example in binding together the representations of SWRs that comprise a train (*Davidson et al., 2009*; *Yamamoto and Tonegawa, 2017*).

Analyses of SWR-associated Ca$^{2+}$ activity also revealed a reduction in Ca$^{2+}$ transient rates prior to SWRs. Small systematic delays in detection of Ca$^{2+}$ signaling relative to electrophysiological events contribute to the estimated timing of Ca$^{2+}$ transient suppression relative to SWRs. Studies linking the action potential to Ca$^{2+}$ transients suggest a lag less than 20 ms based on the rise time constant

(*Jaffe et al., 1992*; *Markram et al., 1995*; *Sah and Clements, 1999*). However, GCaMP6f's rise time constant, ~50 ms at body temperature (*Ohkura et al., 2012*), also contributes. This lag is offset by the ~2–4 ms delay associated with extracellular SWR detection (*Hulse et al., 2016*). Consistent with these expectations, we observed a small (<46 ms) delay between evoked fEPSP and somatic $Ca^{2+}$ fluorescence increases, and in addition observed temporal coincidence of SWRs with neuropil-associated $Ca^{2+}$ activity within the resolution of the 50 ms imaging sampling interval. Accounting for this delay and the offset between $Ca^{2+}$ transient initiation and detection further advances the timing of $Ca^{2+}$ transient reduction relative to SWR onset, suggesting that the SWR-associated reduction in dynamic $Ca^{2+}$ activity does indeed precede SWR onset. Modulation of hippocampal activity by a neocortical DOWN state that precedes SWR events may contribute to the observed suppression of $Ca^{2+}$ transients prior to SWRs (*Sirota et al., 2003*; *Isomura et al., 2006*; *Hahn et al., 2007*; *Hahn et al., 2012*); see also 53,54, as may reduced levels of acetylcholine.

As septal cholinergic neuron activity contributes to theta and reduces SWRs (*Hasselmo, 2006*; *Vandecasteele et al., 2014*; *Norimoto et al., 2012*), we evaluated the contribution of medial septal cholinergic neurons to hippocampal dynamic $Ca^{2+}$ activity. We found that chemogenetic activation of medial septal cholinergic neurons with hM3Dq increased dynamic $Ca^{2+}$ activity while reducing SWR rates. In contrast, inhibition of mAChRs markedly reduced dynamic $Ca^{2+}$ activity while increasing SWRs. Neither chemogenetic activation of cholinergic neurons nor mAChR inhibition altered hippocampal multiunit activity. Although this might be taken as further support of a dissociation between $Ca^{2+}$ activity and action potentials, lack of isolated single units limits interpretation of this observation. Nonetheless, these findings are consistent with reports in reduced preparations showing that mAChRs contribute to somatic $Ca^{2+}$ transients (*Power and Sah, 2002*; *Cho et al., 2008*; *Müller and Connor, 1991*). They demonstrate that medial septal cholinergic activity through activation of mAChRs critically regulates hippocampal dynamic $Ca^{2+}$ activity in vivo, in relation to behavioral and theta state.

Multiple factors may contribute to the apparent dissociation of dynamic $Ca^{2+}$ activity and action potential-associated activity in slow wave sleep. The observed dependence of dynamic $Ca^{2+}$ activity on mAChR activation suggests that low cholinergic tone and thereby reduced mAChR activation may be a critical determinant. In addition, although instantaneous firing rates during theta and SWR events are comparable, differences in spike counts and in patterned activity between these hippocampal oscillations may also contribute. In this respect, it is notable that complex spiking, which has been observed in theta-associated exploratory behavior (*Epsztein et al., 2011*; *Cohen et al., 2017*; *Bittner et al., 2015*; *Bittner et al., 2017*), is particularly effective at driving somatic $Ca^{2+}$ influx (*Grienberger et al., 2014*), while cells typically fire only a few action potentials during SWRs which may be less effective at generating a detectable $Ca^{2+}$ transient. Finally, distinct spatial compartmentalization of $Ca^{2+}$ and differential kinetics and peak concentration of somatic $Ca^{2+}$ transients during exploratory behavior and REM sleep compared to quiet wakefulness and slow wave sleep associated with SWR events may also contribute to the apparent dissociation, and the possibility remains that non-theta states are associated with somatic $Ca^{2+}$ event amplitudes below the resolution of the in vivo microendoscopy technique used here.

ACh plays a critical role in memory formation (*Teles-Grilo Ruivo and Mellor, 2013*; *Hasselmo, 2006*). The high levels of somatic $Ca^{2+}$ activity observed during exploratory behavior-associated theta states that have been implicated in memory encoding (*Buzsáki, 2005*; *Hasselmo, 2006*), coupled with the lack of a robust increase in association with most SWR events implicated in memory consolidation (*Buzsáki, 2015*), suggests that dynamic $Ca^{2+}$ events may contribute preferentially to theta-dependent processes such as memory encoding. As a result, $Ca^{2+}$ dependent gene expression is likely to be enriched in theta-associated exploratory behavior and REM sleep relative to SWR-associated states of quiet wakefulness and slow wave sleep. Even so, the capacity for $Ca^{2+}$ events to coordinate differentially with SWR trains over single SWRs raises the possibility that $Ca^{2+}$ may participate in integrating the representations of extended replay across SWRs. Theta-associated mAChR-dependent cascades that require somatic $Ca^{2+}$ transients may also contribute to hippocampal representations, including place field formation, gain of response (*Cohen et al., 2017*), and stabilization, via both direct effects on ionic conductances as well as effects on $Ca^{2+}$ dependent kinases, phosphatases, and gene expression. NMDA receptor activation (*Silva et al., 2015*) and (intracellular) complex spikes (also referred to as $Ca^{2+}$ plateau potentials; *Epsztein et al., 2011*; *Bittner et al., 2015*; *Bittner et al., 2017*; *Grienberger et al., 2014*; *Takahashi and Magee, 2009*; *Harvey et al., 2009*)

may participate in this process. In contrast, in non-theta states, low levels of mAChR-dependent somatic $Ca^{2+}$ would be anticipated to limit activation of $Ca^{2+}$ dependent intrinsic conductances and second messenger systems, favoring SWR-associated replay and reducing $Ca^{2+}$-dependent gene expression. In the hippocampal slice, activation of mAChRs has been shown to increase intracellular and nuclear $Ca^{2+}$ through Ins-P3 dependent $Ca^{2+}$ release from intracellular stores (*Power and Sah, 2002*; *Cho et al., 2008*). This mechanism may contribute to the actions of ACh in freely behaving animals, driving LTP (*Malenka et al., 1988*; *Dennis et al., 2016*) and $Ca^{2+}$ dependent gene transcription necessary for memory formation (*Bading, 2013*).

The results of this study implicate cholinergic-dependent dynamic $Ca^{2+}$ activity in the molecular cascades necessary for hippocampal memory formation, preferentially during the theta-associated states of exploratory behavior and REM sleep. Impairment of such processes may contribute to hippocampal failure in diseases such as Alzheimer's disease, where loss of cholinergic neurons and $Ca^{2+}$ dyshomeostasis have been described (*Schliebs and Arendt, 2011*; *Magi et al., 2016*; *Berridge, 2014*) and where cholinesterase inhibitors are the only class of drug that consistently improves memory in clinical trials (*Birks et al., 2015*; *Birks, 2006*).

## Materials and methods

All procedures were approved by the Institutional Animal Care and Use Committees of the Massachusetts General Hospital and followed the ethical guidelines of the US National Institutes of Health.

### Preparation for endoscopic calcium imaging and electrophysiology

Adult male C57BL/6J mice underwent injection of 1 μL AAV-hSyn-GCaMP6f-GFP (titer ~$10^{13}$ mL$^{-1}$) into the CA1 region of the hippocampus (anterior-posterior (AP) −2.1 mm, medial-lateral (ML) −1.65 mm, dorsal-ventral (DV) −1.4 mm from Bregma) at 0.2 μL/min under anesthesia (induction and maintenance, isoflurane 0.5–2%), followed by implantation of grin lens above CA1 (DV −1.2 mm from brain surface), as previously described (*Ziv et al., 2013*). A 16-contact linear probe (Neuronexus) or a probe containing eight independently drivable stereotrodes was implanted and lowered into position to sample the ipsilateral CA1 region in the vicinity of the lens, spanning the pyramidal cell layer and including s. radiatum (AP −2.8 mm, ML 2.6 mm from Bregma, −36 ° relative to the AP axis, −39 ° relative to the inter-aural axis; DV −1.7 mm from brain surface). Stereotrodes were adjusted over several days to reach the pyramidal cell layer. Electrodes reliably terminated under the lens. An electrically silent contact, typically the most superficial, served as the reference, with cerebellar ground. For stimulation experiments, an additional stimulus probe made by gluing together a pair of twisted Teflon-coated 90% platinum/10% iridium wires (50 μm inner diameter, 100 μm outer diameter, World Precision Instruments, USA) was implanted into the contralateral CA3 (AP −2.8 mm, ML −3.05 mm from Bregma, −36 ° relative to the AP axis, −39 ° relative to the inter-aural axis; DV −1.5 mm from brain surface). GCaMP6f expression, lens placement, and probe electrode positions were confirmed with postmortem evaluation and histology (*Figure 1—figure supplement 1*).

10 mice received the linear probe; 4 mice received the stereotrode implant. As tabulated below, for assessment of $Ca^{2+}$ activity and LFP across the sleep wake cycle (*Figure 1*), 2 mice had the linear probe and 3 mice had stereotrode implants; for assessment of $Ca^{2+}$ activity in relation to sharp wave ripples (*Figure 2*), 4 mice had the linear probe and 4 mice had stereotrode implants; for assessment of hM3Dq activation (*Figure 3*), 5 mice had the linear probe and no mice had stereotrode implants; for assessment of scopolamine (*Figure 4*), 5 mice had the linear probe and 1 mouse had stereotrode implants. For stimulation experiments, 3 mice received the stereotrode implant in ipsilateral hippocampus with a stimulus probe in contralateral hippocampus (*Figure 2—figure supplement 3*).

### Chemogenetics, cannula implantation and intracranial injection

For chemogenetic experiments, adult *ChAT*-Cre mice (The Jackson laboratory strain B6;129S6-Chat$^{tm1(cre)Low1}$) underwent injection of 0.5 μL AAV-hSyn-DIO-hM3D(Gq)-mCherry (titer ~$10^{13}$ mL$^{-1}$) into the medial septum (AP 0.86 mm, ML −0.5 mm, 6.4° from vertical, DV −4.5 mm from Bregma), as well as 1 μL GCaMP6f virus injection into the hippocampus. For DREADD experiments, animals were first recorded after systemic vehicle injection (i.p, 10 ml/kg in volume) and were then recorded 30 min after clozapine-*N*-oxide injection (CNO, 1 mg/kg i.p), the DREADD ligand. For hM3Dq experiments, one mouse's electrophysiological probe broke.

For intracranial pharmacology experiments, a cannula (26 GA) was placed 1 mm above the dorsal hippocampus, ipsilateral to the miniature microscope (AP −2.8 mm, ML 2.6 mm from Bregma, −36 ° relative to the AP axis, −39 ° relative to the inter-aural axis; DV −0.7 mm from brain surface). Cannula-implanted mice did not undergo implantation of hippocampal electrophysiology probes; instead, skull screws were implanted for EEG for purposes unrelated to the present study. To directly target local hippocampal mAChRs and the cholinergic projection to the hippocampus, animals were first recorded after intrahippocampal injection (i.h., 1 µL in volume, 0.1 µL/min) of vehicle and were then recorded 30 min post 1 µl scopolamine (2 µg/µl, i.h.) or clozapine-N-oxide solution (300 µmol/L, i.h.) injection via the cannula. To assess the accuracy of cannula placement and determine the center of local injection, 0.2 µL trypan blue solution was directly injected into the hippocampus; 1 µL trypan blue injection was used to assess the extent of diffusion (*Figure 3—figure supplement 1*).

## Systemic pharmacological experiments

To test the effects of mAChR inhibition, animals were first recorded after systemic vehicle injection (i.p., 10 ml/kg in volume) and were then recorded 15 min after scopolamine injection (1.5 mg/kg, i.p.). To control for temporal stability of dynamic $Ca^{2+}$ activity, we also assessed recordings in which vehicle was injected both at baseline and in lieu of drug. Animals treated sequentially with vehicle showed stable $Ca^{2+}$ activity during both exploratory behavior and periods of quiet wakefulness (run behavior, $Ca^{2+}$ event rate (mean ± s.e.): vehicle1 0.023 ± 0.005, vehicle2 0.026 ± 0.006, p>0.05, paired t-test; amplitude: vehicle1 3.10 ± 0.05, vehicle2 2.98 ± 0.07, p>0.05, paired t-test; quiet wakefulness, $Ca^{2+}$ event rate (mean ± s.e.): vehicle1 0.016 ± 0.001, vehicle2 0.015 ± 0.001, p>0.05, paired t-test; amplitude: vehicle1 3.25 ± 0.13, vehicle2 3.08 ± 0.09, p>0.05, paired t-test; n = 3).

## Calcium imaging and electrophysiology recording

The Inscopix™ mini-microscope and acquisition system was used for $Ca^{2+}$ imaging. Frame rate was 20 fps (50 ms exposure time). Local field potentials (2 kHz sampling, 0.5–900 Hz filtering) were recorded from each contact site on the probe (Neuralynx™). For multiunit recordings, 1 ms windows around thresholded extracellular action potentials were acquired on-line at 32 kHz, 600–6000 Hz filtering. Head position and direction were monitored with overhead camera tracking of two diodes mounted on the headstage.

Animals were trained over 1–2 weeks to run on a linear track (200 cm) between two goal locations. Electrophysiological recordings were acquired throughout behavioral sessions. $Ca^{2+}$ imaging was acquired simultaneously but in 10 min increments, with 4–6 min periods off-camera to minimize photobleaching. Rest sessions were acquired immediately after behavioral sessions in a sleep chamber within the recording room. Animals were housed in individual cages with a 12 hr light-12h dark standard light cycle.

Exploratory behavior was defined as running behavior (>3 cm/sec) on the linear track. Quiet wakefulness, slow wave sleep, and REM sleep were scored in the sleepbox. Quiet wakefulness was classified during periods of wakeful immobility (<0.5 cm/sec), both prior to sleep and following arousal events and postural changes. Slow wave sleep was classified by irregular activity and SWRs arising in the setting of prolonged immobility (at least 3 min) in a sleep posture. REM sleep was classified by high theta power (5–12 Hz) in the LFP arising out of slow wave sleep in association with persistent immobility, with each epoch of REM lasting at least 10 s. Where available (3/5 animals), EMG confirmed classification.

For stimulation experiments, a Master 8 (AMPI) was used to drive the stimulus isolator (WPI) to produce an input electric current through the (contralateral) stimulating electrode (1 ms duration, 15–20 s interval). This was calibrated to generate field EPSPs on the (ipsilateral) recording stereotrodes which were recorded along with simultaneous calcium imaging. Recordings were performed under isoflurane.

## Data analysis

Established software was used for initial processing of $Ca^{2+}$ activity (Mosaic, Inscopix, Palo Alto, California). This includes $Ca^{2+}$ data registration, using rigid-body motion correction as required, followed by PCA/ICA analysis of ΔF/F datasets, as described previously (*Mukamel et al., 2009*). Matlab (MathWorks, Natick, Massachusetts) was used for further data analysis. ICs were reviewed manually,

and the IC spatial masks were applied to the ΔF/F datasets to derive ΔF/F time series for each soma. For stimulation experiments, the ΔF/F time series were taken from the ICs directly; similar results were obtained using IC spatial masks. Unless otherwise stated, error bars reflect s.e.m.

$Ca^{2+}$ events were identified as threshold crossings of IC traces. Threshold was defined as 3 Z scores above the mean for analyses of dynamic calcium activity across behavior and LFP states. In order to be sensitive to changes in $Ca^{2+}$ event amplitudes with experimental manipulations of the cholinergic system, the threshold for these analyses was taken as 2 Z scores above the mean. Similar results were observed using a threshold of 3 Z scores. The start of each event was defined as the time at which the event reached 20% of its maximum amplitude. To compute the neuropil trace, a region of interest (ROI) was selected that contained all identified somas. From this ROI, we took the maximum intensity of each pixel across the set of normalized ICs to generate a distribution. The set of neuropil pixels was then defined as those pixels lying below the median of this distribution. The neuropil trace was taken as the integral of the ΔF/F time series over the set of neuropil pixels.

Local field potentials across the probe contacts were reviewed and filtered to obtain hippocampal ripples (Blackman filter; 80–400 Hz), theta (5–12 Hz), and delta oscillations (0.5–4 Hz). Channels with strong SWRs and unit activity were taken to be in the pyramidal cell layer and were used for analysis of ripples and multiunit activity. For linear probes, a contact in s. radiatum 100 μm below the cell layer was used for theta and delta. For stereotrode implants, a stereotrode below the cell layer was used for theta and delta. Methods for detecting SWRs and theta were the same for the different types of recordings. SWR events were identified in the sleep chamber and defined using a 3 Z score threshold of ripple power, with the start and end defined as the times at which ripple power returned to the mean, and with a 20 ms minimum duration. Trains of SWRs were identified when the interval between discrete SWRs was less than 1 s. Multiunit activity was selectively analyzed from sites estimated to be in the pyramidal cell layer. Because camera-associated high frequency (1.3 kHz) electrical noise variably contaminated some contacts, action potentials that exceeded threshold were screened on the basis of waveform, and events contaminated with at least 40 μV of 1.3 kHz noise were excluded. Action potentials thus filtered were aggregated as multiunit activity for further analyses. High frequency microscope-associated electrical noise was minimal in the post-behavioral session but was often substantial on the track, prohibiting multiunit contrasts between exploratory behavior and other behavioral states. Spectrograms were assessed from 0 to 20 Hz using a moving window of size 4 s and step size 100 ms, and from 80 to 300 Hz using a moving window of size 100 ms and step size 10 ms, using the Chronux toolbox http://chronux.org/.

$Ca^{2+}$ events associated with SWRs were summed over repeated SWRs to generate peri-event time histograms (PETHs) triggered on the start of SWR events, which were smoothed with a Gaussian window (σ = 50 ms). Baseline $Ca^{2+}$ activity was taken from −5 to −2 s prior to the start of SWR events. Across all SWRs, SWR-associated $Ca^{2+}$ activity was measured at SWR onset. Delayed $Ca^{2+}$ activity was measured 1–2 s after the SWR. For SWR singlet versus train analyses, SWR-associated $Ca^{2+}$ activity was measured at SWR offset, and delayed $Ca^{2+}$ activity was measured within 3 s of the SWR in a 500 ms window centered around the maximum.

Synchronous $Ca^{2+}$ events (SCE) were identified when the number of cells with $Ca^{2+}$ event onsets within a 250 ms window exceeded the number expected by chance, as defined by >3 s.d. after 1000 temporal shuffles of cell activity, and with a minimum of 5 participating cells (*Malvache et al., 2016*). The onset time of the first transient within the SCE was used to relate SCEs to ripples. A co-occurrence was identified when SCEs started within −100/+150 ms of ripples. Significant ripple-associated SCEs were identified when their ripple power exceeded the median + twice the interquartile range of the distribution of ripple power associated with 1000 temporal shuffles of SCE times.

For place field assessment, spatial tuning curves (3 cm bins) were constructed for each running direction using all detected $Ca^{2+}$ events that occurred during run behavior (>3 cm/sec). Spatial information was measured for each spatial tuning curve and compared to spatial information distributions of 1000 shuffled versions of the data to test significance (Monte Carlo p-value<0.05) (*Markus et al., 1994*). In each shuffle, $Ca^{2+}$ events were reassigned to random times and spatial information was recomputed.

## Statistical analysis

All group values were represented as mean ± s.e.m. Student paired *t*-test and one-way ANOVA with repeated measures followed by *post hoc* tests were used for statistical analyses according to experimental designs (SPSS 13.0 version) (*Supplementary file 1*). The significance level was set at $p < 0.05$.

## Histology

Animals were injected with Fatal plus (100 mg/kg, i.p.) and perfused with 4% paraformaldehyde solution. Probe and lens locations were assessed (*Figure 1—figure supplement 1*). Brains were removed and cut in coronal sections with a cryostat (50 µm-thickness). Expression of hM3Dq was assessed using mCherry antibody (ab167453, 1:500, Abcam), along with anti-ChAT antibody (AB144P, 1:100, Millipore). Fluorescence expression and electrode location were evaluated with microscopy (Zeiss Axio Imager Z2, Germany).

## Acknowledgements

This work was supported by R01 AG054551 and Fidelity Biosciences awards.

## Additional information

### Funding

| Funder | Grant reference number | Author |
|---|---|---|
| Fidelity Biosciences | | Bradley T Hyman |
| Fidelity Biosciences | | Stephen N Gomperts |
| National Institute on Aging | 1R01 AG054551 | Stephen N Gomperts |

The funders had no role in study design, data collection and interpretation, or the decision to submit the work for publication.

### Author contributions

Heng Zhou, Formal analysis, Validation, Investigation, Methodology, Writing—review and editing; Kevin R Neville, Conceptualization, Software, Formal analysis, Investigation, Visualization, Methodology, Writing—review and editing; Nitsan Goldstein, Data curation, Investigation, Writing—review and editing; Shushi Kabu, Naila Kausar, Data curation, Software, Investigation, Writing—review and editing; Rong Ye, Thuan Tinh Nguyen, Investigation, Writing—review and editing; Noah Gelwan, Software, Validation, Writing—review and editing; Bradley T Hyman, Resources, Funding acquisition, Writing—review and editing; Stephen N Gomperts, Conceptualization, Resources, Data curation, Software, Formal analysis, Supervision, Funding acquisition, Validation, Investigation, Visualization, Methodology, Writing—original draft, Project administration, Writing—review and editing

### Author ORCIDs

Heng Zhou (iD) http://orcid.org/0000-0001-8187-6124
Stephen N Gomperts (iD) https://orcid.org/0000-0002-0083-0077

### Ethics

Animal experimentation: This study was performed in strict accordance with the recommendations in the Guide for the Care and Use of Laboratory Animals of the National Institutes of Health. All of the animals were handled according to an approved institutional animal care and use committee (IACUC) protocol (2012N000206) of the Massachusetts General Hospital. All surgery was performed under isoflurane anesthesia, and every effort was made to minimize suffering.

### Decision letter and Author response

Decision letter https://doi.org/10.7554/eLife.39777.034
Author response https://doi.org/10.7554/eLife.39777.035

## Additional files

### Supplementary files

• Supplementary file 1. Statistical analyses for group data.
DOI: https://doi.org/10.7554/eLife.39777.029

• Transparent reporting form
DOI: https://doi.org/10.7554/eLife.39777.030

### Data availability

Imaging data has been deposited into Dryad, and is available at doi:10.5061/dryad.8ct101p

The following dataset was generated:

| Author(s) | Year | Dataset title | Dataset URL | Database and Identifier |
|---|---|---|---|---|
| Zhou H, Neville K, Goldstein N | 2019 | Data from: Cholinergic modulation of hippocampal calcium activity across the sleep-wake cycle | https://doi.org/10.5061/dryad.8ct101p | Dryad Digital Repository, 10.5061/dryad.8ct101p |

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
