## [Decision Letter]

Thank you for sending your article entitled "Cholinergic modulation of hippocampal calcium activity across the sleep-wake cycle" for peer review at *eLife*. Your article has been reviewed by three peer reviewers, one of whom is a member of our Board of Reviewing Editors, and the evaluation has been overseen by Michael Frank as the Senior Editor.

All reviewers agreed that your study addresses important questions, namely how calcium dynamics depend on behavioral states, how calcium activity relates to hippocampal network states (i.e., theta vs. sharp wave-ripples), and how imaged calcium activity is affected by cholinergic neuromodulatory inputs from the medial septum. All reviewers agreed that your paper potentially shows important and interesting results and uses an impressive combination of techniques. However, reviewers also agreed that the conclusions are not convincingly supported by the results as currently described. A major concern is a lack of control animals receiving CNO injections in the absence of DREADD expression. Another major concern is whether the timing of calcium transients and sharp wave-ripples could be compared with sufficient temporal resolution in order to convincingly conclude that calcium activity decreases during extracellularly recorded sharp wave-ripples.

The essential revisions are listed below, roughly in order of importance, and these are the points that should be addressed. These essential revisions are followed by the three reviews in their entirety, for your information.

Essential revisions:

1) An important control is missing from these experiments. The authors did not examine the effects of CNO alone, without hM3Dq receptor expression (Gomez et al., Nature 2017). CNO can also bind to other endogenous receptors. Thus, controls with only CNO injected (in absence of DREADDs) are needed to determine whether the reported changes in calcium activity in the chemogenetic manipulation experiments are not due to effects caused by CNO itself.

2) The possibility that transients are suppressed during SWRs could be very interesting, but there was insufficient accuracy and confidence in the temporal resolution to draw any conclusions. It is difficult to determine the exact onset of a calcium transient at the timescales necessary for any firm conclusions. Furthermore, one cannot be certain whether or not there is a systematic lag between electrophysiological and calcium events in the recordings. So, it remains possible that neuronal firing during SWRs produces calcium transients that are observed slightly after LFP-detected ripples.

3) How exactly were exploratory behaviors, slow wave sleep, and REM sleep identified and defined? The authors report that these states were associated with different theta/δ ratios, but theta/δ ratios are typically used to differentiate REM sleep from slow wave sleep, thus making this "result" a consequence of the method. Also, the theta/δ ratio in Figure 1 does not show a significant increase in the run versus rest behavioral state. This is important for the paper, and the authors should explain this apparently anomalous result. Moreover, the authors state in the Results section that "As expected, both exploratory behavior and REM sleep were associated with a high theta/δ ratio, which diminished in quiet wakefulness and slow wave sleep (repeated measures ANOVA: F (3, 12) =3.965, p>0.05; Figure 1I)"; However, their statistical results do not seem to support this statement. The authors should show group-averaged or individual power spectrum analysis of theta power changes with behavior states along with theta/δ ratios and clarify these puzzling results.

4) Activation of either cholinergic cells in MS or their axonal inputs to hippocampus, reportedly leads to increased rates of calcium events during quiet wakefulness. However, even though both i.p and i.h injections of CNO to activate either the cholinergic cells in MS or cholinergic inputs to hippocampus, respectively, were reportedly done during quiet wakefulness, the running velocities in Figure 3L and Figure 3T are very different. Why is this? Also, running speeds greater than 5 cm/s, as shown in Figure 3L, are not typically considered quiet wakefulness. Similarly, mAChR inhibition carried out by either i.p or i.h injection in Figure 4 both resulted in decreases in the rate of calcium events. However, the measured running velocities shown in Figure 4H and 4P are significantly different. In which behavioral states were these manipulations carried out?

5) Since the central point of this manuscript is a link between behavioral states and calcium dynamics, it would be crucial to provide evidence of the behavioral state in which each of the manipulations were carried out to evoke changes in calcium dynamics. Thus, it would be useful to have pre-injection and post-injection comparisons of behavioral state and calcium dynamics. The measures and statistics shown are insufficient in this regard.

6) Related to the above point, the corresponding increase or decrease of theta power upon activation or inhibition of the cholinergic system would be useful to correlate with the various manipulations and calcium activity changes.

7) The details about electrophysiological recordings are unclear. The authors state that they used a 16-contact linear probe in some animals, or 8 independently drivable stereotrodes in other animals. It is unclear how many mice had each kind of recording. Also, which channels were used for analyses? Also, were the methods for detecting sharp wave-ripples and theta the same for the different types of recordings?

8) Related to the above point, the authors also state that a skull screw was used to obtain EEG recordings in animals with cannula. How were the authors certain that sharp wave-ripples and theta could be detected from a skull screw? The authors need to provide example recordings to demonstrate that these patterns could be detected.

9) Was multiunit activity recorded from all depths/recording sites of the probes or just those sites estimated to be in stratum pyramidale? This is important to assess the likelihood that multiunit recordings included mostly interneuron spikes or more likely picked up mostly pyramidal cell activity.

10) Related to the above point, MUA activity should be shown alongside the calcium transients throughout the figures in the paper.

11) The reported silence in calcium transients prior to SWRs is interesting, but seems to follow from prior observations by Sirota et al., 2003 and Isomura et al., 2006, where it was noted that SWRs are frequently preceded by DOWN states, which are in turn also known to modulate hippocampal activity (see also Hahn et al., 2007 and 2012).

12) The authors should include a Statistical Analysis section in the Materials and methods part with details of the tests used. The authors should also provide exact p-values, statistics, and associated degrees of freedom for all results, including non-significant results (i.e., not just p > 0.5).

13) In Video 1, why was the background so much higher during RUN and were comparisons of calcium activity across states affected by these differences?

14) The experiment with CNO activation of MS cholinergic cells and then mAChR inhibition (Supplementary Figure 4) is very important to show that calcium activity increases with cholinergic activation is really mediated through mAChR. It should be in the main figures to support a major finding of the paper, namely the role of mAChR in Ca^2+^ activity.

15) Why do some results refer to 5 mice and others refer to 6?

The separate reviews follow.

*Reviewer #1:*

This is an interesting study that uses simultaneous calcium imaging and electrophysiological recording of neuronal activity in CA1 of freely behaving mice to test how calcium activity differs across different states (i.e., theta and sharp wave-ripple-related states) and how such dynamics are affected by cholinergic projections from the medial septum. Although my enthusiasm for this unique study is high, the presentation of results is lacking in several regards. My specific comments are as follows:

1) How were exploratory behaviors, slow wave sleep, and REM sleep identified and defined? The authors report that these states were associated with different theta/δ ratios, but theta/δ ratios are typically used to differentiate REM sleep from slow wave sleep.

2) The authors should provide exact p-values, statistics, and associated degrees of freedom for all results, including non-significant results (i.e., not just p > 0.5).

3) The details about electrophysiological recordings are unclear. The authors state that they used a 16-contact linear probe in some animals, or 8 independently drivable stereotrodes in other animals. It is unclear how many mice had each kind of recording. Also, which channels were used for analyses? Also, were the methods for detecting sharp wave-ripples and theta the same for the different types of recordings?

4) Related to the above point, was multiunit activity recorded from all depths of the probes or just from stratum pyramidale? This is important regarding whether some of the multiunit recordings likely included interneurons or more likely picked up pyramidal cells.

5) Also related to the above point, the authors also state that a skull screw was used to obtain EEG recordings in animals with cannula. How were the authors certain that sharp wave-ripples and theta could be detected from a skull screw? The authors need to provide example recordings to demonstrate that these patterns could be detected.

6) Why do some results refer to 5 mice and others refer to 6?

7) In Video 1, why was the background so much higher during RUN and how did this affect comparisons of calcium activity across states?

*Reviewer #2:*

This manuscript shows that hippocampal dynamic calcium activity depends on both the behavioral states of the mice and on the function of the cholinergic neuromodulatory system. In recent years, several studies have demonstrated the usefulness and importance of somatic Ca^2+^ dynamics in behaving animals in real time studies of neuronal activity. Thus, it is important to understand the relation between this calcium activity and local field potentials during different behavioral states, and how this calcium activity is modulated. The studies presented in this manuscript demonstrate that behavioral states affect the dynamics of Ca^2+^ activity of neuronal ensembles in the hippocampus in a mAChR dependent manner. Behavioral states that engaged the theta rhythm, like exploration and REM sleep, were characterized by higher rates and amplitudes of Ca^2+^ events, in contrast to behavioral states characterized by low levels of theta and higher levels of SWR, such as quite wakefulness (rest) and slow wave sleep. Furthermore, cholinergic inputs from medial septum, as well as endogenous mAChR activation in hippocampus, were shown to modulate dynamic calcium activity in hippocampal networks. These results are interesting and important, but the authors should address the issues outlined below:

a) Figure 1: The theta/δ ratio in the figure does not show a significant increase in the run versus rest behavioral state. This is important for the paper and the authors should explain this apparently anomalous result. Moreover, the authors state in the Results section that "As expected, both exploratory behavior and REM sleep were associated with a high theta/δ ratio, which diminished in quiet wakefulness and slow wave sleep (repeated measures ANOVA: F (3, 12) =3.965, p>0.05; Figure 1I)"; However, their results do not seem to support this statement. They should show group-averaged or individual power spectrum analysis of the theta power changes with behavior states, instead of, or along with, the theta/δ ratios.

b) Activation of either cholinergic cells in MS or their axonal inputs to hippocampus, leads to increased rates of Ca^2+^ event during quite wakefulness. However, even though both i.p and i.h injections of CNO to activate either the cholinergic cells in MS or cholinergic inputs to hippocampus, respectively, were done during quite wakefulness (rest) state, the running velocities in Figure 3L and Figure 3T are very different. Why is this?

c) Similarly, mAChR inhibition carried out by either i.p or i.h during exploratory behavior (run) state in Figure 4, both result in decreases in the rate of Ca^2+^ events. However, the results of the measured running velocities shown in Figure 4H and 4P are significantly different.

d) Since the central point of this manuscript is a link between behavioral states and Ca^2+^ dynamics, it would be crucial to provide evidence of the behavioral state in which each of the manipulations were carried out to evoke changes in Ca^2+^ dynamics. Thus, it would be useful to have pre-injection and post-injection comparisons.

e) Similarly, the corresponding increase or decrease of theta power upon activation or inhibition of the cholinergic system would be useful to correlate with the various manipulations and Ca^2+^ dynamic changes.

f) The experiment with CNO activation of MS cholinergic cells and then mAChR inhibition (Supplementary Figure 4) is very important to show that the Ca^2+^ event increase with cholinergic activation is really mediated through mAChR. It should be in the main figures to support the major finding of the paper: the role of mAChR in Ca^2+^ dynamic activity. Additional controls with only CNO injected (in absence of DREADDs) are also needed to determine whether the changes in Ca^2+^ are actually due to changes in mAChR function and not to CNO itself.

g) Please include a Statistical Analysis section in the Materials and methods part with details of the tests used.

In summary: the premise of the studies presented is important and interesting. However, the authors need to address the issues listed above before their paper is ready for publication.

*Reviewer #3:*

In this work, Zhou and colleagues perform recordings that feature an impressive combination of calcium imaging, lfp and multi-unit extracellular recordings, chemogenetics, and pharmacology. They make a series of interesting observations: that calcium transients are suppressed prior to the onset of hippocampal sharp-wave ripples (SWRs), that injections of CNO activation in mice with septal Chat-hM3DQq receptors increases calcium transients, and that scopolamine injections blocked calcium transients. These observations are used to argue that muscarinic acetylcholine receptors play a critical role in hippocampal calcium activity. If convincingly presented, these are important findings as they point towards dissociations between calcium and extracellular activities and a specific role for mAchRs. However, I had concerns with the approach for each of the findings and at present I am not convinced that confounds cannot account for the various observations.

The silence in calcium transients prior to SWRs is interesting, but seems to follow from prior observations by Sirota et al., 2003 and Isomura et al., 2006, where it was noted that SWRs are frequently preceded by DOWN states, which are in turn also known to modulate hippocampal activity (see also Hahn et al., 2007 and 2012).

*The possibility that transients are suppressed during* SWRs themselves could be very interesting, but there was insufficient accuracy and confidence in the temporal resolution to draw any conclusions here. It is difficult to determine the exact onset of a calcium transient at the timescales necessary for any firm conclusions. Furthermore, one cannot be certain that there isn't a systematic lag between electrophysiological and calcium events in the recordings. So from what I can tell, it remains possible that neuronal firing during SWRs produces calcium transients that are observed slightly after the lfp-detected ripple.

The CNO hM3Dq results are interesting, but an important control is missing from these experiments. The authors did not examine the effects of CNO alone, without the hM3Dq receptor expression (Gomez et al. Nature 2017) – CNO can also bind to other endogenous receptors. Effect of CNO with hM4DI could have provided a compelling contrast, but these were not performed.

In these and elsewhere it would be important that the MUA activity is shown alongside the Ca transients throughout the figures in the paper.

[Editors' note: further revisions were requested prior to acceptance, as described below.]

Thank you for resubmitting your work entitled "Cholinergic modulation of hippocampal calcium activity across the sleep-wake cycle" for further consideration at *eLife*. Your revised article has been favorably evaluated by Michael Frank as the Senior Editor, and three reviewers, one of whom is a member of our Board of Reviewing Editors.

The manuscript has been improved but there are some remaining issues that need to be addressed before acceptance, as outlined below:

*Reviewer #3:*

The revision by Zhou et al. is certainly an improvement, but I have remaining concerns.

Analyses in Figure 2 are based on different windows, "PRE" "SWR" and "POST", but it is not clear what time periods these correspond to and whether those periods are appropriate for the different measures. The legends indicate that PRE is 2-5 s before the SWR and POST is 1-2 s after the SWR. Perhaps -2 to +1 s around the SWR are considered the SWR period, but I could not be sure. In any case, it's not clear whether these periods are appropriate for the different analyses because the Ca rate and the Neuropil z-score have different dynamics. Ca rate drops in the ~1s before the SWR, while Neuropil peaks in the ~1s after the SWR. Using the same "SWR" windows for both analyses may obscure significant changes, potentially why the increase in Neuropil does not reach significance. The relevant time periods should be easily identified from the figure and its legends.

I am still confused by what the "Neuropil" z-score is really measuring. I had recommended a discussion of this measure, which the authors did not address. The only thing I can find about this is from the Materials and methods that "from this ROI the 50% of pixels with below-average variance in fluorescence were used." It's not clear to me what exactly this is measuring. As defined, some of these pixels may in fact overlap with parts of somata. Is that appropriate? Is the neuropil also corrected for motion? I believe the authors should discuss and provide a rationale for what they believe this measure provides. The ROI and some sample "neuropil" pixels should be shown. Importantly, do the remaining 50% of pixels behave consistently with the neuropil pixels? It would also be helpful to illustrate the non-thresholded Ca activity for the somata used for event detection, as it remains possible that calcium events amplitudes are lower during SWRs and therefore that the transients are generally being missed. This possibility should be considered in the Discussion section as cells typically show only a few action potentials during SWRs, perhaps not enough to produce a sufficiently large Ca transient.

I had also indicated previously, but that authors did not regard the Niethard et al., 2016 study. I still think it's important for the authors to cite and discuss this paper, in which they recorded Ca activity in the cortex of mice across awake and sleep states. Unlike the current study, they found the least Ca activity during REM, except in PV cells, with intermediate levels during SWS. Therefore, state-dependent and cholinergic effects may not be uniform across brain regions.

---

## [Author Response]

To address the request to evaluate LFP in relation to population calcium activity, we explicitly assessed the LFP associated with synchronous calcium events across neurons. Synchronous events were relatively rare (10 ± 6 per recording, n=5 animals). Consistent with the report of Malvache et al., 12.6 ± 5.6% of synchronous calcium events time locked to SWRs. Furthermore, compared to temporally shuffled data, 4.9 ± 3.1% of synchronous calcium activities were associated with increased ripple power (p<0.05, n=5 animals). These data confirm the capacity for synchronous calcium events to coincide with SWRs. (These data are discussed in the fifth paragraph of the subsection “Coordination of hippocampal dynamic calcium activity across behavior”.)

To address the concern of a systematic delay between electrophysiological measurements and calcium measurements, we first accounted for the delay between calcium transient detection, defined as the time at which the event reached 20% of its maximum amplitude, and the start of calcium transients on average (0.114 ± 0.001 s; n=7 mice). Accounting for this delay, the maximal reduction in calcium activity preceded SWR onset by 0.189 ± 0.118 sec. Consistent with the capacity for calcium imaging to detect the SWR-associated synaptic barrage with temporal fidelity, as noted previously, SWRs were associated with a time-locked (mean ± s.e.: -0.04 ± 0.10 sec) change in the extra-somatic calcium fluorescence of the neuropil, with a nonsignificant increase followed by a significant, delayed reduction (repeated measures ANOVA: F (2, 12) =15.627, p=0.003, n=7 animals; post-hoc comparisons: Pre vs. SWR, p=0.130; Pre vs. Post, p<0.001; SWR vs. Post, p=0.003; Figure 2G and K, left).

To independently assess the temporal alignment of the LFP and the calcium fluorescence data, we placed a stimulating electrode in contralateral CA3 to activate the commissural pathway and evoke both a field EPSP and an associated change in GCaMP6f fluorescence in CA1 neurons. Stimulation evoked a field EPSP with mean peak latency 20.8 ± 0.7 ms and an associated increase in somatic calcium fluorescence that started within 66.7 ± 16.7 ms (1.3 ± 0.3 samples) and peaked within 100 ± 0 ms (2 ± 0 samples; n = 3 animals; Figure 2—figure supplement 2). Accounting for this offset and the ~190 ms offset delineated above further advances the timing of dynamic calcium activity relative to SWR onset. Together, these results support the observation that dynamic calcium activity transiently falls in the immediate pre-SWR period, consistent with a down-state in the pre-SWR period, with a subsequent increase in calcium activity elicited by SWR trains. (We have provided these results in the third paragraph of the aforementioned subsection.)

Essential revisions:1) An important control is missing from these experiments. The authors did not examine the effects of CNO alone, without hM3Dq receptor expression (Gomez et al., Nature 2017). CNO can also bind to other endogenous receptors. Thus, controls with only CNO injected (in absence of DREADDs) are needed to determine whether the reported changes in calcium activity in the chemogenetic manipulation experiments are not due to effects caused by CNO itself.

To evaluate for CNO off-target effects, CNO was first injected i.p. in animals lacking hM3Dq receptor expression. No effect was observed on the rate of dynamic Calcium activity (Calcium event rates, p=0.496; amplitudes, p=0.710; paired t-test, n=5; Figure 3W and Figure 3—figure supplement 4). In addition, intra-hippocampal injection of CNO in animals lacking hM3Dq receptor expression did not alter on the rate of dynamic calcium activity (calcium event rates, p=0.637; amplitudes, p=0.128; paired t-test, n=4; Figure 3W and Figure 3—figure supplement 4).

2) The possibility that transients are suppressed during SWRs could be very interesting, but there was insufficient accuracy and confidence in the temporal resolution to draw any conclusions. It is difficult to determine the exact onset of a calcium transient at the timescales necessary for any firm conclusions. Furthermore, one cannot be certain whether or not there is a systematic lag between electrophysiological and calcium events in the recordings. So, it remains possible that neuronal firing during SWRs produces calcium transients that are observed slightly after LFP-detected ripples.

The SWR-triggered reduction in dynamic calcium activity started and was often greatest before the SWR began (minimum, mean ± s.e.: -0.075 ± 0.118 sec). To refine the temporal relationship between calcium activity and SWRs, as noted above, we first accounted for the delay between calcium transient detection, defined as the time at which the event reached 20% of its maximum amplitude, and the start of calcium transients on average (0.114 ± 0.001 s; n=7 mice). Accounting for this delay, the maximal reduction in calcium activity preceded SWR onset by 0.189 ± 0.118 sec. Consistent with the capacity for calcium imaging to detect the SWR-associated synaptic barrage with temporal fidelity, SWRs were associated with a time-locked (mean ± s.e.: -0.04 ± 0.10 sec) change in the extra-somatic calcium fluorescence of the neuropil, with a nonsignificant increase followed by a significant, delayed reduction (repeated measures ANOVA: F _(2, 12)_ =15.627, p=0.003, n=7 animals; post-hoc comparisons: Pre *vs.* SWR, p=0.130; Pre *vs.* Post, p<0.001; SWR *vs.* Post, p=0.003; Figure 2G and K, left). To independently assess the temporal alignment of the LFP and the calcium fluorescence data, we placed a stimulating electrode in contralateral CA3 to activate the commissural pathway and evoke both a field EPSP and an associated change in GCaMP6f fluorescence in CA1 neurons. Stimulation evoked a field EPSP with mean latency 20.8 ± 0.7 ms and an associated increase in somatic calcium fluorescence that started within 66.7 ± 16.7 ms (1.3 ± 0.3 samples) and peaked within 100 ± 0 ms (2 ± 0 samples; n = 3 animals; Figure 2—figure supplement 2). Accounting for this offset and the ~190 ms offset delineated above further advances the timing of dynamic calcium activity relative to SWR onset. Together, these results support the observation that dynamic calcium activity transiently falls in the immediate pre-SWR period. (subsection “Coordination of hippocampal dynamic calcium activity across behavior”, third paragraph).

3) How exactly were exploratory behaviors, slow wave sleep, and REM sleep identified and defined? The authors report that these states were associated with different theta/δ ratios, but theta/δ ratios are typically used to differentiate REM sleep from slow wave sleep, thus making this "result" a consequence of the method.

We have clarified these definitions of behavioral state in the manuscript (Materials and methods, subsection “Calcium imaging and electrophysiology recording”). Exploratory behavior was defined as running behavior (>3 cm/sec) on the linear track. Quiet wakefulness, slow wave sleep, and REM sleep were scored in the sleepbox. Quiet wakefulness was classified during periods of wakeful immobility (< 0.5 cm/sec), both prior to sleep and following arousal events and postural changes. Slow wave sleep was classified by irregular activity and SWRs arising in the setting of prolonged immobility (at least 3 minutes) in a sleep posture. REM sleep was classified by high theta power (5-12 Hz) in the LFP arising out of slow wave sleep in association with persistent immobility, with each epoch of REM lasting at least 10 seconds. Where available (3/5 animals), EMG confirmed classification. We have modified Figure 1L to contrast theta across behavioral states excluding REM sleep.

Also, the theta/δ ratio in Figure 1 does not show a significant increase in the run versus rest behavioral state. This is important for the paper, and the authors should explain this apparently anomalous result.

We have modified Figure 1L (subsection “Coordination of hippocampal dynamic calcium activity across behavior”, first paragraph) to contrast theta across behavioral states, excluding REM sleep. As expected, there was a significant main effect of behavioral state on theta power (repeated measures ANOVA excluding REM: F (2,8) =20.468, p=0.001), with higher theta power in exploratory behavior than quiet wakefulness and SWS (post hoc comparisons with run: quiet wakefulness, p=0.003; slow wave sleep, p=0.015; Figure 1L).

Moreover, the authors state in the Results section that "As expected, both exploratory behavior and REM sleep were associated with a high theta/δ ratio, which diminished in quiet wakefulness and slow wave sleep (repeated measures ANOVA: F (3, 12) =3.965, p>0.05; Figure 1I)"; However, their statistical results do not seem to support this statement.

We now state: “As expected, there was a significant main effect of behavioral state on theta power (repeated measures ANOVA excluding REM: F _(2,8)_ =20.468, p=0.001), with higher theta power in exploratory behavior than quiet wakefulness and SWS (post hoc comparisons with run: quiet wakefulness, p=0.003; slow wave sleep, p=0.015; Figure 1L).”

The authors should show group-averaged or individual power spectrum analysis of theta power changes with behavior states along with theta/δ ratios and clarify these puzzling results.

Figure 1L shows group-averaged theta power as a function of behavioral state.

4) Activation of either cholinergic cells in MS or their axonal inputs to hippocampus, reportedly leads to increased rates of calcium events during quiet wakefulness. However, even though both i.p and i.h injections of CNO to activate either the cholinergic cells in MS or cholinergic inputs to hippocampus, respectively, were reportedly done during quiet wakefulness, the running velocities in Figure 3L and Figure 3T are very different. Why is this?

We now report velocities in quiet wakefulness (Figure 3M, V) for direct comparison of i.p. and i.h. CNO injections.

Also, running speeds greater than 5 cm/s, as shown in Figure 3L, are not typically considered quiet wakefulness.

As noted above, we now report velocities in quiet wakefulness (Figure 3M, V) for direct comparison of i.p. and i.h. CNO injections.

Similarly, mAChR inhibition carried out by either i.p or i.h injection in Figure 4 both resulted in decreases in the rate of calcium events. However, the measured running velocities shown in Figure 4H and 4P are significantly different. In which behavioral states were these manipulations carried out?

As noted above, systemic injection of scopolamine was studied during run behavior, where we would anticipate the greatest effect of scopolamine on calcium activity. Figure 4J, K reflect this. Intrahippocampal injection of scopolamine was studied in quiet wakefulness (Figure 4S), because the implantation of the intracranial cannula reduced stability of calcium imaging during running. Figure 4—figure supplement 2 shows the impact on scopolamine i.p. on velocity in the sleep box.

5) Since the central point of this manuscript is a link between behavioral states and calcium dynamics, it would be crucial to provide evidence of the behavioral state in which each of the manipulations were carried out to evoke changes in calcium dynamics. Thus, it would be useful to have pre-injection and post-injection comparisons of behavioral state and calcium dynamics. The measures and statistics shown are insufficient in this regard.

We have modified the manuscript to clarify that animals were awake during all manipulations, to report time spent moving in the sleepbox, and in an independent set of mice, we have reported the effect of CNO and scopolamine injections on detailed behavior (Figure 3—figure supplement 2, Figure 4—figure supplement 1).

6) Related to the above point, the corresponding increase or decrease of theta power upon activation or inhibition of the cholinergic system would be useful to correlate with the various manipulations and calcium activity changes.

We now report that neither CNO-associated changes in SWR rates (p=0.4, Spearman) nor changes in theta power (p=0.7, Spearman) significantly correlated with CNO-associated changes in calcium event rates. In addition, we report that neither scopolamine-associated changes in SWR rates (p=0.3, Spearman) nor changes in theta power (p=0.14, Spearman) significantly correlated with changes in calcium event rates. (subsection “Role for acetylcholine in hippocampal neuron dynamic calcium activity”).

7) The details about electrophysiological recordings are unclear. The authors state that they used a 16-contact linear probe in some animals, or 8 independently drivable stereotrodes in other animals. It is unclear how many mice had each kind of recording. Also, which channels were used for analyses? Also, were the methods for detecting sharp wave-ripples and theta the same for the different types of recordings?

We have clarified these details in the Materials and methods (probes are detailed in the subsection “Preparation for endoscopic calcium imaging and electrophysiology”; channels and methods for theta and SWR detection in the subsection “Data analysis”).

8) Related to the above point, the authors also state that a skull screw was used to obtain EEG recordings in animals with cannula. How were the authors certain that sharp wave-ripples and theta could be detected from a skull screw? The authors need to provide example recordings to demonstrate that these patterns could be detected.

We have clarified that skull screws were implanted for EEG in cannula-implanted mice for purposes unrelated to the present study (subsection “Chemogenetics, cannula implantation and intracranial injection”).

9) Was multiunit activity recorded from all depths/recording sites of the probes or just those sites estimated to be in stratum pyramidale? This is important to assess the likelihood that multiunit recordings included mostly interneuron spikes or more likely picked up mostly pyramidal cell activity.

We state that multiunit activity was selectively analyzed from sites estimated to be in the pyramidal cell layer (subsection “Data analysis”).

10) Related to the above point, MUA activity should be shown alongside the calcium transients throughout the figures in the paper.

We now show MUA in each figure.

11) The reported silence in calcium transients prior to SWRs is interesting, but seems to follow from prior observations by Sirota et al., 2003 and Isomura et al., 2006, where it was noted that SWRs are frequently preceded by DOWN states, which are in turn also known to modulate hippocampal activity (see also Hahn et al., 2007 and 2012).

We agree and as noted above have pursued additional experiments and analyses which confirm that calcium transient reduction precedes SWR onset, consistent with the possibility that this reduction may reflect modulation of hippocampal activity by this neocortical DOWN state.

12) The authors should include a Statistical Analysis section in the Materials and methods part with details of the tests used. The authors should also provide exact p-values, statistics, and associated degrees of freedom for all results, including non-significant results (i.e., not just p > 0.5).

We have done so.

13) In Video 1, why was the background so much higher during RUN and were comparisons of calcium activity across states affected by these differences?

We used identical display settings across the videos to highlight the significant differences in calcium activity across behavioral states. The higher background observed during run behavior is likely to reflect a combination of extra-somatic calcium activity in the neuropil and out of plane somatic calcium activity. The PCA/ICA approach used is resilient to this issue, and we focused on changes in calcium fluorescence. As a result, comparisons of somatic calcium activity across states are insensitive to differences in background.

14) The experiment with CNO activation of MS cholinergic cells and then mAChR inhibition (Supplementary Figure 4) is very important to show that calcium activity increases with cholinergic activation is really mediated through mAChR. It should be in the main figures to support a major finding of the paper, namely the role of mAChR in Ca^2+^ activity.

We have incorporated these data into Figure 3 (Figure 3X, Y).

15) Why do some results refer to 5 mice and others refer to 6?

We have clarified this in the Materials and methods (subsection “Preparation for endoscopic calcium imaging and electrophysiology”).

[Editors' note: further revisions were requested prior to acceptance, as described below.]

The manuscript has been improved but there are some remaining issues that need to be addressed before acceptance, as outlined below:

Reviewer #3:

The revision by Zhou et al. is certainly an improvement, but I have remaining concerns.Analyses in Figure 2 are based on different windows, "PRE" "SWR" and "POST", but it is not clear what time periods these correspond to and whether those periods are appropriate for the different measures. The legends indicate that PRE is 2-5 s before the SWR and POST is 1-2 s after the SWR. Perhaps -2 to +1 s around the SWR are considered the SWR period, but I could not be sure.

Thank you for this point. We apologize for any confusion. As stated in the Materials and methods, “Baseline Ca^2+^ activity was taken from -5 to -2 seconds prior to the start of SWR events. Across all SWRs, SWR-associated Ca^2+^ activity was measured at SWR onset. Delayed Ca^2+^ activity was measured 1-2 seconds after the SWR.” These points were also noted in the legend for Figure 2. Thus, Ca^2+^ activity was measured instantaneously at the time of SWR onset. We have clarified this further in the figure and legend.

In any case, it's not clear whether these periods are appropriate for the different analyses because the Ca rate and the Neuropil z-score have different dynamics. Ca rate drops in the ~1s before the SWR, while Neuropil peaks in the ~1s after the SWR. Using the same "SWR" windows for both analyses may obscure significant changes, potentially why the increase in Neuropil does not reach significance. The relevant time periods should be easily identified from the figure and its legends.

We agree that Ca^2+^ dynamics in the somas and the neuropil are distinct. However, as inspection of Figure 2 shows, and as discussed in the Results, the minimum of somatic Ca^2+^ rate and the maximum of neuropil Ca^2+^ activity both occur at approximately the time of SWR onset (0 s). As requested, we have added the time periods to the figure to complement this information in the legend.

I am still confused by what the "Neuropil" z-score is really measuring. I had recommended a discussion of this measure, which the authors did not address. The only thing I can find about this is from the Materials and methods that "from this ROI the 50% of pixels with below-average variance in fluorescence were used." It's not clear to me what exactly this is measuring.

Thank you for this helpful comment. To define the set of neuropil pixels, we first took the maximum intensity of each pixel across the set of normalized ICs to generate a distribution. The set of neuropil pixels was then defined as those pixels lying below the median of this distribution. The neuropil trace was taken as the integral of the ΔF/F time series over the set of neuropil pixels. We have clarified this in the Materials and methods (subsection “Data analysis”).

As defined, some of these pixels may in fact overlap with parts of somata. Is that appropriate? Is the neuropil also corrected for motion?

In practice, ICs that had low spatial variance and ICs that were not spatially compact were discarded. As a result, the neuropil measure as defined above effectively excludes the pixels that meaningfully contribute to somas. Please note that the entire video undergoes motion correction as an early pre-processing step, as noted in the subsection “Data analysis”, including all pixels contributing to the neuropil measure and to somas.

I believe the authors should discuss and provide a rationale for what they believe this measure provides. The ROI and some sample "neuropil" pixels should be shown.

We have increased discussion and rationale for the use of this measure (Results subsection “Coordination of hippocampal dynamic calcium activity across behavior”, third paragraph) and have provided a figure (Figure 2—figure supplement 2). Because GCaMP6f expression is present in both somas and dendrites (e.g., Chen et al., 2013), we anticipated that an aggregate of extrasomatic pixels would enrich for dendritic calcium dynamics capable of reflecting SWR-associated synaptic activity with temporal fidelity.

Importantly, do the remaining 50% of pixels behave consistently with the neuropil pixels?

Based on the definition of the neuropil measure, the remaining 50% of pixels will include pixels contributing to identified somas, pixels contributing to unidentified somas, and pixels contributing to extrasomatic processes. As this is a complex measure of uncertain biological meaning, we have not studied it.

It would also be helpful to illustrate the non-thresholded Ca activity for the somata used for event detection, as it remains possible that calcium events amplitudes are lower during SWRs and therefore that the transients are generally being missed. This possibility should be considered in the Discussion section as cells typically show only a few action potentials during SWRs, perhaps not enough to produce a sufficiently large Ca transient.

Figure 2A illustrates the non-thresholded Ca activity used for event detection. We provide an additional plot below. We previously noted in the Discussion that “although instantaneous firing rates during theta and SWR events are comparable, differences in spike counts and in patterned activity between these hippocampal oscillations may also contribute”, and that “the possibility remains that non-theta states are associated with somatic Ca^2+^ event amplitudes below the resolution of the in vivo microendoscopy technique used here“. We now have expanded this Discussion as requested (fifth paragraph).
